

**Comparisons of stemflow yield and efficiency between two xerophytic**
**shrubs: the effects of leaves and implications in drought tolerance**
**C. Yuan**[1,2], **G. Y. Gao**[1,3], **B. J. Fu**[1,3]
[1] State Key Laboratory of Urban and Regional Ecology, Research Center for
Eco-Environmental Sciences, Chinese Academy of Sciences, Beijing 100085, China
[2] University of Chinese Academy of Sciences, Beijing 100049, China
[3] Joint Center for Global Change Studies, Beijing 100875, China
*Correspondence to*: **G. Y. Gao** (gygao@rcees.ac.cn)
Tel.: +86 10 62841239





**Abstract.**
Stemflow transports enriched precipitation to the rhizosphere and is highly important for
the survival of xerophytic shrubs in water-stressed ecosystems. However, its ecological
significance has generally been underestimated because it is relatively limited in amount, and
the biotic mechanisms that affect it have not been thoroughly studied at the leaf scale. In this
study, the branch stemflow volume ($SF_b$), the shrub stemflow equivalent water depth ($SF_d$),
the stemflow percentage of incident precipitation (SF%), the stemflow productivity (SFP),
the funnelling ratio (FR), the rainfall characteristics and the plant traits of branches and
leaves of *C. korshinskii* and *S. psammophila* were measured during the 2014 and 2015 rainy
seasons in the northern Loess Plateau of China. This study evaluated the stemflow production
efficiency for the first time with the combined results of SFP and FR, and sought to determine
the inter- and intra-specific differences in stemflow production and production efficiency, as
well as the specific bio-/abiotic mechanisms that affected stemflow. The results indicated that
precipitation amount was the most influential rainfall characteristic that affected stemflow in
these two endemic shrub species and that stem biomass and leaf biomass were the most
influential plant traits in *C. korshinskii* and *S. psammophila*, respectively. *C. korshinskii* had a
greater stemflow production and production efficiency at all precipitation levels, and the
largest inter-specific difference was generally in the 5–10-mm young shoots during the most
frequent rainfall events of ≤2 mm. *C. korshinskii* had a lower precipitation threshold (0.9 mm
vs. 2.1 mm for *S. psammophila*), which provided more available water from rainfall for
stemflow. The leaves affected stemflow production, and the beneficial leaf traits contributed
to the higher stemflow production of *C. korshinskii*. In summary, *C. korshinskii* might have
greater drought tolerance and a competitive edge in a dryland ecosystem because of greater
and more efficient stemflow production, a lower precipitation threshold and more
advantageous leaf traits.
*Keywords*: Xerophytic shrub; Stemflow production; stemflow production efficiency;
Threshold precipitation; Beneficial leaf traits.



## 1 Introduction

Stemflow channels divert precipitation pointedly into the root zone of a plant via
preferential root paths, worm paths and soil macropores. The double-funnelling effects of
stemflow and preferential flow create "hot spots" and "hot moments" by enhancing
biogeochemical reactivity at the terrestrial-aquatic interface (McClain et al., 2003; Johnson
and Lehmann, 2006), thus substantially contributing to the formation and maintenance of
so-called "fertile islands" (Whitford et al., 1997), "resource islands" (Reynolds et al., 1999)
or "hydrologic islands" (Rango et al., 2006). This effect is important for the normal function
of rain-fed dryland ecosystems (Wang et al., 2011).
Shrubs are a representative plant functional type (PFT) in dryland ecosystems and have
developed effective physiological drought tolerance by reducing water loss, e.g., through
adjusting their photosynthetic and transpiration rate by regulating stomatal conductance and
abscisic acid (ABA), titling their osmotic equilibrium by regulating the concentration of
soluble sugars and inorganic ions, and removing free radicals (Ma et al., 2004, 2008). The
efficient production of stemflow is a vital eco-hydrological flux involved in soil water
replenishment (Pressland 1973) as well as an effective strategy to acquire water (Murakami,
2009) and withstand drought (Martinez-Meza and Whitford, 1996). However, because
stemflow occurs in small amounts, previous studies have usually ignored stemflow (Llorens
and Domingo, 2007) and have underestimated its disproportionately high influence on the
survival and competitiveness of xerophytic shrub species. Therefore, the quantification of
inter- and intra-specific stemflow production is important to assess the stemflow production
efficiency and to elucidate the underlying bio-/abiotic mechanisms.
Stemflow production includes the stemflow volume and depth, and it describes the total
flux channelled down to the base of a branch or a trunk, but stemflow data are unavailable for
comparison of inter-specific differences caused by variations in the branch architecture, the



canopy structure, the shrub species and the eco-zone. Herwitz (1986) introduced the
funnelling ratio (FR), which is expressed as the quotient of the volume of stemflow produced
and the product of the base area and the precipitation amount. It indicates the efficiency with
which individual branches or shrubs capture raindrops and deliver the water to the root zone
(Siegert and Levia, 2014). The FR allows a comparison of the inter- and intra-specific
stemflow production under different precipitation conditions. However, the FR does not
provide a connection between hydrological processes (e.g., rainfall redistribution) and the
plant growth processes (e.g., biomass accumulation and allocation). Recently, Yuan et al.
(2016) have introduced the parameter stemflow productivity (SFP), expressed as the volume
of stemflow production per unit of branch biomass. The SFP describes the efficiency in an
energy-conservation manner by comparing the stemflow volume of a unit biomass increment
of different-sized branches.
The precipitation amount is an abiotic mechanism that has been recognized as the single
most influential rainfall characteristic (Clements 1972; André et al., 2008; Van Stan et al.,
2014). However, in terms of biotic mechanisms, although the canopy structure (Mauchamp
and Janeau, 1993; Crockford and Richardson, 2000; Pypker et al., 2011) and branch
architecture (Herwitz, 1987; Murakami 2009; (Herwitz, 1987;Murakami, 2009;
Carlyle-Moses and Schooling, 2015) have been studied for years, the most important plant
traits that vary with location and shrub species have not yet been determined. The effects of
the leaves have been studied more recently at a smaller scale, e.g., leaf orientation (Crockford
and Richardson, 2000), shape (Xu et al., 2005), arrangement pattern (Owens et al., 2006),
pubescence (Garcia-Estringana et al., 2010), area (Sellin et al., 2012), epidermis microrelief
(Roth-Nebelsick et al., 2012), amount (Li and Xiao, 2016), biomass (Yuan et al., 2016; Li et
al., 2016), etc. Although comparisons of stemflow production during the foliated and dormant
seasons usually indicate negative effects of leaves because the more stemflow occurred at the





leafless period (Dolman, 1987; Neal et al., 1993; Mużyło et al., 2012), both negligible and
positive effects have also been confirmed by Martinez-Meza and Whitford (1996) and Liang
et al. (2009), respectively. Nevertheless, the validity of these findings has been called into
question as a result of the seasonal variation of meteorological conditions and plant traits, e.g.,
wind speed (André et al., 2008), rainfall intensity (Dunkerley et al., 2014 a, b), air
temperature and consequent precipitation type (snow-to-rain vs. snow) (Levia, 2004).
Therefore, a controlled experiment with foliated and manually defoliated plants under the
same stand conditions is needed to resolve these uncertainties.
In this study, the branch stemflow volume ($SF_b$), the shrub stemflow depth ($SF_d$), the
stemflow percentage of the incident precipitation amount (SF%), the SFP and the FR were
measured in two shrub species (*C. korshinskii* and *S. psammophila*) endemic to a semiarid
area of northern China during the 2014 and 2015 rainy seasons. The objectives of this study
were to (1) quantify the inter- and intra-specific stemflow production ($SF_b$, $SF_d$ and SF%) and
the production efficiency (SFP and FR); (2) investigate the effects of the rainfall
characteristics and plant traits on the stemflow in these two shrub species; and (3) specifically
identify leaf characteristics that affect the stemflow with respect to morphology, structural
characteristics and the biomass partitioning pattern. The achievement of these research
objectives would provide a novel characterization of plant drought tolerance and species
competitiveness in terms of stemflow and further the understanding of the effects of leaves on
the survival and growth of plants from an eco-hydrological perspective.

**2 Materials and Methods**
**2.1 Study area**
This study was conducted at the Liudaogou catchment (110°21′-110°23′E,
38°46′-38°51′N) in Shenmu County in the Shaanxi Province of China. It is 6.89 km$^2$ and





1094-1273 m above sea level (a.s.l.). This area has a semiarid continental climate with
well-defined rainy and dry seasons. The mean annual precipitation (MAP) between 1971 and
2013 was 414 mm, with approximately 77% of the annual precipitation amount occurring
during the rainy season (Jia et al., 2013), which lasts from July to September. The mean
annual temperature and potential evaporation are 9.0 ℃ and 1337 mm year$^{-1}$ (Zhao and Shao,
2009), respectively. The coldest and warmest months are January and July, with an average
monthly temperature of 9.7 ℃ and 23.7 ℃, respectively. Two soil types of Aeolian sandy soil
and Ust-Sandiic Entisol dominate this catchment (Jia et al., 2011). Soil particles consist of
11.2%-14.3% clay, 30.1%-44.5% silt and 45.4%-50.9% sand in terms of the soil classification
system of United States Department of Agriculture (Zhu and Shao, 2008). The original plants
are scarcely present, except for very few surviving shrub species, e.g., *Ulmus macrocarpa*,
*Xanthoceras sorbifolia*, *Rosa xanthina*, *Spiraea salicifolia*, etc. The currently predominant
shrub species were planted decades ago, e.g., *S. psammophila*, *C. Korshinskii*, *Amorpha*
*fruticosa*, etc., and the predominant grass species include *Medicago sativa*, *Stipa bungeana*,
*Artemisia capillaris*, *Artemisia sacrorum*, etc. (Ai et al., 2015).
*C. Korshinskii* and *S. psammophila* are endemic shrub species in arid and semiarid
northern China and were planted for wind-proofing and dune-stabilizing because of their
great drought tolerance. Two representative experimental stands were established in the
southwest of the Liudaogou catchment (Fig. 1). Both *C. korshinskii* and *S. psammophila* were
planted approximately twenty years ago, and the two stands share a similar slope of 13-18 °, a
size of 3294-4056 m$^2$, and an elevation of 1179-1207 m a.s.l. However, the *C. korshinskii*
experimental stand had a 224 ° aspect with a loess ground surface, whereas the *S.*
*psammophila* experimental stand had a 113 °aspect with a sand ground surface.

Fig. 1. Location of the experimental stands and facilities for stemflow measurements of *C.*
*korshinskii* and *S. psammophila* at the Liudaogou catchment in the Loess Plateau of China.



**2.2 Field experiments**
Field experiments were conducted during the rainy seasons of 2014 (July 1 to October 3)
and 2015 (June 1 to September 30) to measure the rainfall characteristics, plant traits and
stemflow. To avoid the effects of gully micro-geomorphology on recording the rainfall
characteristics, we installed an Onset® (Onset Computer Corp., Bourne, MA, USA) RG3-M
tipping bucket rain gauge (0.2 mm per tip) at each experimental stand. Three 20-cm-diameter
rain gauges were placed around to adjust the inherent underestimating of automatic
precipitation recording (Groisman and Legates, 1994). Then, rainfall duration (RD, h),
rainfall interval (RI, h), the average rainfall intensity (I, mm h$^{-1}$), the maximum rainfall
intensity in 5 min ($I_5$, mm h$^{-1}$), 10 min ($I_{10}$, mm h$^{-1}$) and 30 min ($I_{30}$, mm h$^{-1}$) could be
calculated accordingly. In this study, the individual rainfall events were greater than 0.2 mm
and separated by a period of at least four hours without rain (Giacomin and Trucchi, 1992).
*C. korshinskii* and *S. psammophila*, as modular organisms and multi-stemmed shrub
species, have branches of that exist as independent individuals. Therefore, we focused on the
inter- and intra-specific branch stemflow by experimenting on sample shrubs that had a
similar canopy structure. Four mature shrubs were selected for *C. korshinskii* (designated as
C1, C2, C3 and C4) and *S. psammophila* (designated as S1, S2, S3 and S4) for the stemflow
measurements. They had isolated canopies, similar intra-specific heights and canopy areas,
e.g., 2.1 ± 0.2 m and 5.14 ± 0.26 m$^2$ for C1-C4, and 3.5 ± 0.2 m and 21.35 ± 5.21 m$^2$ for
S1-S4. We measured the morphological characteristics of all the 180 branches of C1-C4 and
all the 261 branches of S1-S4, including the branch basal diameter (BD, mm), branch length
(BL, cm) and branch inclination angle (BA, º). The leaf area index (LAI) and the foliage
orientation (MTA, the mean tilt angle of leaves) were measured using LiCor® (LiCor
Biosciences Inc., Lincoln, NE, USA) 2200C plant canopy analyser approximately twice a
month.





A total of 53 branches of *C. korshinskii* and 98 branches of *S. psammophila* were
selected for stemflow measurements following the criteria: 1) no intercrossing stems; 2) no
turning point in height from branch tip to the base; 3) representativeness in amount and
branch size. Stemflow was collected using aluminum foil collars, which was fitted around the
entire branch circumference and sealed by neutral silicone caulking (Fig. 1). A
0.5-cm-diameter PVC hose led the stemflow to lidded containers. The stemflow volume was
measured within two hours after the rainfall ended during the daytime; if the rainfall ended at
night, we took the measurement early the next morning.
Another three shrubs of each species were destructively measured for biomass and leaf
traits. They had similar canopy heights and areas as those of the shrubs for which the
stemflow was measured and were designated as C5-C7 (2.0-2.1 m and 5.84-6.77 m$^2$) and
S5-S7 (3.0-3.4 m and 15.43-19.20 m$^2$), thus allowing the development of allometric models
for the estimation of the corresponding biomass and leaf traits of C1-C4 and S1-S4 (Levia
and Herwitz, 2005; Siles et al., 2010a, 2010b; Stephenson et al., 2014). A total of 66 branches
for C5-C7 and 61 branches for S5-S7 were measured when the shrubs showed maximum
vegetative growth during mid-August for the biomass of leaves and stems (BML and BMS,
g), the leaf area of the branches (LAB, cm$^2$), and the leaf numbers of the branches (LNB).
The BML and BMS were weighted after oven-drying of 48 hours. The detailed measurements
have been reported in Yuan et al., (2016). The validity of the allometric models was verified
by measuring another 13 branches of C5-C7 and 14 branches of S5-S7.

**188    2.3 Calculations**

Biomass and leaf traits were estimated by allometric models as an exponential function
of BD (Siles et al., 2010a, b; Jonard et al., 2006):
$$PT_e = a * BD^b \qquad (1)$$




where $a$ and $b$ are constants, and $PT_e$ refers to the estimated plant traits BML, BMS, LAB
and LNB. The other plant traits could be calculated accordingly, including individual leaf
area of branch (ILAB = 100*LAB/LNB, $mm^2$), the percentage of stem biomass to that of
branch (PBMS = BMS/(BML+BMS)*100%, %), specific leaf weight (SLW = BML/LAB,
$g\ cm^{-2}$), Huber value (HV = BBA/LAB = $3.14*BD^2/(400*LAB)$, unitless, where BBA is the
branch basal area ($cm^2$)).

In this study, stemflow production was defined as the branch volume production

(hereafter "stemflow production", $SF_b$, mL), the equivalent water depth on the basis of shrub
canopy area (hereafter "stemflow depth", $SF_d$, mm), and the stemflow percentage of the
incident precipitation amount (hereafter "stemflow percentage", SF%, %):
$$SF_d = 10 * \sum\nolimits_{i=1}^{n} SF_{b_i}/CA \tag{2}$$

$$SF\% = (SF_d/P)*100\% \tag{3}$$

where $SF_{bi}$ is the volume of stemflow production of branch $i$ (mL), CA is the canopy area
($cm^2$), n is the number of branches, and P is the incident precipitation amount (mm).

Stemflow productivity (SFP, $mL\ g^{-1}$) was expressed as the $SF_b$ (mL) of unit branch

biomass (g) and represented the stemflow production efficiency of different-sized branches in
terms of energy-conservation:
$$SFP = SF_b/(BML + BMS) \tag{4}$$

The funnelling ratio (FR) was computed as the quotient of $SF_b$ and the product of P and

BBA (Herwitz, 1986). A FR with a value greater than 1 indicated a positive effect of the
canopy on the stemflow production (Carlyle-Moses and Price, 2006). The value of (P * BBA)
equals to the precipitation amount that would have been caught by the rain gauge occupying
the same basal area at the clearing:
$$FR = 10*SF_b/(P*BBA) \tag{5}$$






**2.4 Data analysis**

A Pearson correlation analysis was performed to test the relationship between $SF_b$ and each of the rainfall characteristics and plant traits. Significantly correlated variables were further tested with a partial correlation analysis for their separate effects on $SF_b$. Then, the qualified variables were fed into a stepwise regression with forward selection to identify the most influential bio-/abiotic factors (Carlyle-Moses and Schooling, 2015; Yuan et al., 2016). Similarly to a principal component analysis and ridge regression, stepwise regression has commonly been used because it gets a limited effect of multicollinearity (Návar and Bryan, 1990; Honda et al., 2015; Carlyle-Moses and Schooling, 2015). Moreover, we excluded variables that had a variance inflation factor (VIF) greater than 10 to minimize the effects of multicollinearity (O'Brien, 2007). The same analysis method was also applied to identify the most influential bio-/abiotic factors affecting SFP and FR. The level of significance was set at 95% confidence interval ($p = 0.05$). The SPSS 20.0 (IBM Corporation, Armonk, NY, USA), Origin 8.5 (OriginLab Corporation, Northampton, MA, USA), and Excel 2013 (Microsoft Corporation, Redmond, WA, USA) were used for data analysis.

**3 Results**

**3.1 Species-specific variation of plant traits**

According to the *Flora of China* and the field observation, both *C. korshinskii* and *S. psammophila* had an inverted-cone canopy and no trunk, with the branches running obliquely from the base. *S. psammophila* usually grew to 3-4 m and had an odd number of strip-shaped leaves of 24-mm in width and 4080-mm in length. The young leaves were pubescent and gradually became subglabrous (Chao and Gong, 1999) (Fig. 2). In comparison, *C. korshinskii* usually grew to 2 m and had pinnate compound leaves with 12-16 foliates in an opposite or sub-opposite arrangement (Wang et al., 2013). The leaf was concave and lanceolate-shaped,





with an acute leaf apex and an obtuse base. Both sides of the leaves were densely sericeous
with appressed hairs (Liu et al., 2010) (Fig. 2).

Fig. 2. Comparison of leaf morphologies of *C. korshinskii* and *S. psammophila*.

Allometric models were developed to estimate the biomass and leaf traits of the

branches of *C. korshinskii* and *S. psammophila* measured for stemflow. The quality of the
estimates was verified by linear regression. As shown in Fig. 3, the regression of LAB, LNB,
BML and BMS of *C. korshinskii* had an approximately 1:1 slope (0.99 for the biomass
indicators and 1.04 for the leaf traits) and an $R^2$ value of 0.93-0.95. According to Yuan et al.,
(2016), the regression of *S. psammophila* had a slope of 1.13 and an $R^2$ of 0.92. Therefore,
those allometric models were appropriate.

Fig. 3. Verification of the allometric models for estimating the biomass and leaf traits of *C.*
*korshinskii*. BML and BMS refer to the biomass of the leaves and stems, respectively, and
LAB and LNB refer to the leaf area and the number of branches, respectively.

*C. korshinskii* had a similar average branch size and angle, but a shorter branch length

than did *S. psammophila*, e.g., 12.48 ± 4.16 mm vs. 13.73 ± 4.36 mm, 60 ± 18 ° vs. 60 ± 20 °,
and 161 ± 35 cm vs. 267.3 ± 49.7 cm, respectively. Regarding branch biomass accumulation,
*C. korshinskii* had a smaller BML (an average of 19.93 ± 10.81 g) and a larger BMS (an
average 141.07 ± 110.78 g) than did *S. psammophila* (an average of 27.85 ± 20.71 g and
130.65 ± 101.35 g, respectively). Both the BML and BMS increased with increasing branch
size for these two shrub species. When expressed as a proportion, *C. korshinskii* had a larger
PBMS than that of *S. psammophila* in all the BD categories. The PBMS-specific difference
increased with an increasing branch size, ranging from 1.24% for the 5–10-mm branches to
7.22% for the >18-mm branches.

Although an increase in LAB and LNB and a decrease in ILAB were observed for both



shrub species with an increase in branch size, *C. korshinskii* had a larger LAB (an average of
2509.05 ± 1355.30 cm$^2$) and LNB (an average of 12479 ± 8409), but a smaller ILAB (an
average of 21.94 ± 2.99 mm$^2$) than did *S. psammophila* for each BD level (Table 1). The
inter-specific differences in the leaf traits decreased with increasing branch size. The largest
difference occurred for the 5–10-mm branches, e.g., LNB and LAB were 12.21-fold and
2.41-fold larger for *C. korshinskii*, and ILAB was 5.32-fold larger for *S. psammophila*. *C.*
*korshinskii* had a larger SLW (an average of 126.04 ± 0.29 g cm$^{-2}$) and HV (0.0507 ± 0.0064)
than did *S. psammophila* (73.87 ± 14.52 g cm$^{-2}$ and 0.0009 ± 0.0001, respectively). As the
branch size increased, the SLW of *S. psammophila* decreased from 95.62 g cm$^{-2}$ for the 5–
10-mm branches to 58.07 g cm$^{-2}$ for the >18-mm branches, but the HV of *C. korshinskii*
increased from 0.0438 to 0.0615.
Table 1. Comparison of branch morphology, biomass and leaf traits of *C. korshinskii* and *S.*
*psammophila*.

**3.2 Stemflow production of *C. korshinskii* and *S. psammophila***

In this study, stemflow production was expressed as $SF_b$ on the branch scale and $SF_d$ and

SF% on the shrub scale. The $SF_b$ was an average of 290.6 mL and 150.3 mL for individual
branches of *C. korshinskii* and *S. psammophila*, respectively. The $SF_b$ was positively
correlated with the branch size and precipitation of these two shrub species. As the branch
size increased, $SF_b$ increased from 119.0 mL for the 5–10-mm branches to 679.9 mL for
the >20-mm branches for *C. korshinskii* and from 43.0 mL to 281.8 mL for the corresponding
BD categories of *S. psammophila*. However, with increasing precipitation, a larger
intra-specific difference in $SF_b$ was observed, which increased from 28.4 mL during rains ≤2
mm to 771.4 mL during rains >20 mm for *C. korshinskii* and from 9.0 mL to 444.3 mL for
the corresponding precipitation categories of *S. psammophila*. The intra-specific differences
in $SF_b$ were significantly affected by the rainfall characteristics and the plant traits. Up to



2375.9 mL of stemflow was measured for the >18-mm branches of *C. korshinskii* during
rains >20 mm, but only 6.8 mL of stemflow occurred for the 5–10-mm branches during rains
≤2 mm. For comparison, a maximum $SF_b$ of 2097.6 mL and a minimum of 1.8 mL were
measured for *S. psammophila*.
*C. korshinskii* produced a larger $SF_b$ than did *S. psammophila* for all BD and
precipitation categories, and the inter-specific differences in $SF_b$ also varied substantially
with the rainfall characteristics and the plant traits. A maximum difference of 4.3-fold larger
for the $SF_b$ of *C. korshinskii* was observed for the >18-mm branches during rains ≤2 mm. As
the precipitation increased, the $SF_b$-specific difference decreased from 3.2-fold larger for *C.*
*korshinskii* during rains ≤2 mm to 1.7-fold larger during rains >20 mm. The largest
$SF_b$-specific difference occurred for the 5–10-mm branches for almost all precipitation
categories, but no clear trend of change was observed with increasing branch size (Table 2).
$SF_d$ and SF% averaged 1.00 mm and 8.0%, respectively, for individual *C. korshinskii*
shrubs and 0.8 mm and 5.5%, respectively, for individual *S. psammophila* shrubs. These
parameters increased with increasing precipitation, ranging from 0.09 mm and 5.8% during
rains ≤2 mm to 2.64 mm and 8.9% during rains >20 mm for *C. korshinskii* and from 0.01 mm
and 0.7% to 2.23 mm and 7.9% for the corresponding precipitation categories of *S.*
*psammophila*, respectively. Additionally, the individual *C. korshinskii* shrubs had a larger
stemflow than did *S. psammophila* for all precipitation categories. The maximum differences
in $SF_d$ and SF% were 8.5- and 8.3-fold larger for *C. korshinskii* during rains ≤2 mm and
decreased with increasing precipitation to 1.2- and 1.1-fold larger during rains >20 mm.

Table 2. Comparison of stemflow production ($SF_b$, $SF_d$ and SF%) between *C. korshinskii* and
*S. psammophila*.

**3.3 Stemflow production efficiency of *C. korshinskii* and *S. psammophila***
Combined results for SFP and FR, the stemflow production efficiency were assessed for





*C. korshinskii* and *S. psammophila*. SFP averaged 1.95 mL g$^{-1}$ and 1.19 mL g$^{-1}$ for individual
*C. korshinskii* and *S. psammophila* branches, respectively (Table 3). As precipitation
increased, SFP increased from 0.19 mL g$^{-1}$ during rains ≤2 mm to 5.08 mL g$^{-1}$ during
rains >20 mm for *C. korshinskii* and from 0.07 mL g$^{-1}$ to 3.43 mL g$^{-1}$ for the corresponding
precipitation categories for *S. psammophila*. With an increase in branch size, SFP decreased
from 2.19 mL g$^{-1}$ for the 5–10-mm branches to 1.62 mL g$^{-1}$ for the >18-mm branches of *C.*
*korshinskii* and from 1.64 mL g$^{-1}$ to 0.80 mL g$^{-1}$ for the corresponding BD categories of *S.*
*psammophila*. Maximum SFP values of 5.60 mL g$^{-1}$ and 4.59 mL g$^{-1}$ were recorded for *C.*
*korshinskii* and *S. psammophila*, respectively. Additionally, *C. korshinskii* had a larger SFP
than that of *S. psammophila* for all precipitation and BD categories. This inter-specific
difference in SFP decreased with increasing precipitation from 2.5-fold larger for *C.*
*korshinskii* during rains ≤2 mm to 1.5-fold larger during rains >20 mm, and it increased with
increasing branch size: from 1.3-fold larger for *C. korshinskii* for the 5–10-mm branches to
2.0-fold larger for the >18-mm branches.
Table 3. Comparison of stemflow productivity (SFP) between *C. korshinskii* and *S.*
*psammophila*.

FR averaged 172.3 and 69.3 for the individual branches of *C. korshinskii* and *S.*
*psammophila*, respectively (Table 4). As the precipitation increased, an increasing trend was
observed, ranging from 129.2 during rains ≤2 mm to 190.3 during rains >20 mm for *C.*
*korshinskii* and from 36.7 to 96.1 during the corresponding precipitation categories for *S.*
*psammophila*. FR increased with increasing BA from 149.9 for the ≤30º-branches to 198.2
for the >80 º branches of *C. korshinskii* and from 55.0 to 85.6 for the corresponding BA
categories of *S. psammophila*. Maximum FR values of 276.0 and 115.7 were recorded for *C.*
*korshinskii* and *S. psammophila*, respectively. Additionally, *C. korshinskii* had a larger FR
than *S. psammophila* for all precipitation and BA categories. The inter-specific difference in





FR decreased with increasing precipitation from the 3.5-fold larger for *C. korshinskii* during
rains ≤2 mm to 2.0-fold larger during rains >20 mm, and it decreased with an increase in the
branch inclination angle: from 2.7-fold larger for *C. korshinskii* for the ≤30º-branches to
2.3-fold larger for the >80 ºbranches.
Table 4. Comparison of the funnelling ratio (FR) for *C. korshinskii* and *S. psammophila*.

**3.4 Bio/abiotic influential factors of stemflow production and production efficiency**
For both *C. korshinskii* and *S. psammophila*, BA was the only plant trait that had no
significant correlation with $SF_b$ ($r < 0.13$, $p > 0.05$) as indicated by Pearson correlation
analysis. The separate effects of the remaining plant traits were verified by using a partial
correlation analysis, but BL, ILAB and PBMS failed this test. The remaining plant traits,
including BD, LAB, LNB, BML and BMS, were regressed with $SF_b$ by using the forward
selection method. Biomass was finally identified as the most important biotic indicator that
affected stemflow, which behaved differently in *C. korshinskii* for BMS and in *S.*
*psammophila* for BML. The same analysis methods indicated that the precipitation amount
was the most important rainfall characteristic that affected stemflow in these two shrub
species.
$SF_b$ and $SF_d$ had a good linear relationship with the precipitation amount ($R^2 \geq 0.93$) for
both shrub species (Fig. 4). The >0.9-mm and >2.1-mm rains were required to start $SF_b$ for *C.*
*korshinskii* and *S. psammophila*, respectively, results consistent with the 0.8-mm and 2.0-mm
precipitation threshold calculated with $SF_d$. Moreover, the precipitation threshold increased
with increasing branch size. The precipitation threshold values were 0.69 mm, 0.72 mm, 1.35
mm and 0.81 mm for the 5–10-mm, 10–15-mm, 15–18-mm and >18-mm branches of *C.*
*korshinskii*, respectively, and 1.1 mm, 1.6 mm, 2.0 mm and 2.4 mm for the branches of *S.*
*psammophila*, respectively.



The SF% of the two shrub species also increased with precipitation, but was inversely
proportional and gradually approached asymptotic values of 9.1% and 7.7% for *C.*
*korshinskii* and *S. psammophila*, respectively. As shown in Fig. 4, fast growth was evident
during rains ≤10 mm, but SF% slightly increased afterwards for both shrub species.
Fig. 4. Relationships of branch stemflow production ($SF_b$), shrub stemflow depth ($SF_d$) and
stemflow percentage (SF%) with precipitation amount (P) for *C. korshinskii* and *S.*
*psammophila*.

Precipitation amount was the most important factor affecting SFP and FR for *C.*
*korshinskii* and *S. psammophila*, but the most important biotic factor was different. BA was
the most influential plant trait that affected FR, and ILAB was the most important plant trait
affecting SFP during rains ≤10 mm. However, during heavy rain, BD and PBMS were the
most significant biotic factors for *C. korshinskii* and *S. psammophila*, respectively.

**4 Discussion**
**4.1 Effective utilization of precipitation via stemflow production**
Stemflow in *C. korshinskii* and *S. psammophila* increased with increasing precipitation
and branch size at both the branch ($SF_b$) and shrub scales ($SF_d$ and SF%). However, *C.*
*korshinskii* had larger $SF_b$, $SF_d$ and SF% values than did *S. psammophila* for all precipitation
categories. Although the greatest stemflow production was observed during rains >20 mm for
the two shrub species, the inter-specific differences of $SF_b$, $SF_d$ and SF% were highest at 3.2-,
8.5- and 8.3-fold larger for *C. korshinskii* during rains ≤2 mm, which indicated that *C.*
*korshinskii* utilized precipitation far more effectively during rains ≤2 mm at the branch and
shrub scale. These data indicate that stemflow was highly important for the survival of the
xerophytic shrubs in extreme drought. Additionally, *C. korshinskii* had a 2.8-fold larger $SF_b$
than that of *S. psammophila* for the 5–10-mm branches. Therefore, compared with *S.*




*psammophila*, more effectively might *C. korshinskii* utilize precipitation via greater stemflow
production, particularly the 5–10-mm young shoots during rains ≤2 mm.
The FR values indicated the efficiency with which individual branches could intercept
and channel raindrops (Siegert and Levia, 2014), thus leading to greater stemflow production.
The average FR of *S. psammophila* was 69.3, which agreed well with the 69.4 of *S.*
*psammophila* in the Mu Us sandland in China (Yang et al., 2008). The average FR for *C.*
*korshinskii* was 173.3, in contrast to the values of 156.1 (Jian et al., 2014) and 153.5 (Li et al.,
2008) for *C. korshinskii* in the western Loess Plateau of China. Furthermore, these two shrub
species had a larger FR than those of many other endemic xerophytic shrubs from
water-stressed ecosystems, e.g., *Tamarix ramosissima* (24.8) (Li et al., 2008), *Artemisia*
*sphaerocephala* (41.5) (Yang et al., 2008), *Reaumuria soongorica* (53.2) (Li et al., 2008),
*Hippophae rhamnoides* (62.2) (Jian et al., 2014). Therefore, both *C. korshinskii* and *S.*
*psammophila* utilized precipitation in a relatively efficient manner by producing stemflow,
and *C. korshinskii* produced stemflow more efficiently. The FR-specific difference achieved a
maximum of 3.5-fold larger for *C. korshinskii* during rains ≤2 mm and decreased with
increasing precipitation to 2.0-fold larger during rains >20 mm.
SFP characterized stemflow production in terms of energy-conservation. *C. korshinskii*
had a larger SFP than *S. psammophila* for all the precipitation and BD categories, and during
rains ≤2 mm, the SFP-specific difference was maximized to 2.5-fold larger for *C. korshinskii*.
Additionally, the 5–10-mm branches had the largest average SFP of 2.2 mL g$^{-1}$ and 1.6
mL g$^{-1}$ in return, which, during rains >20 mm, was maximized to 5.6 mL g$^{-1}$ and 4.6 mL g$^{-1}$
for *C. korshinskii* and *S. psammophila*, respectively (Table 3). Investing biomass into young
shoots provides considerable water benefits for xerophytic shrubs. Therefore, compared with
*S. psammophila*, more efficiently might *C. korshinskii* utilize precipitation by producing
greater stemflow, particularly for 5–10-mm young shoots during rains ≤2 mm.





Stemflow may preferentially incorporate precipitation into the rhizosphere, retaining it as relatively stable soil moisture (Martinez-Meza and Whitford, 1996) and increasing drought tolerance, particularly during long periods without rain. It was particularly significant that young shoots were favoured in the presence of a greater water supply. Greater stemflow production provided *C. korshinskii* with greater drought tolerance and a competitive edge in water-stressed ecosystems.

**4.2 Utilization of more rains via a low precipitation threshold to start stemflow**

Precipitation below the threshold wet the canopy and then evaporated, so it did not generate stemflow. The ≤2.5-mm rains were entirely intercepted and evaporated to the atmosphere for the xerophytic Ashe juniper communities at the central Texas of USA (Owens et al., 2006), as well as most of the ≤5-mm rains, particularly at the beginning raining stage for xerophytic shrubs (*S. psammophila*, *Hedysarum scoparium*, *A. sphaerocephala* and *Artemisia ordosica*) at the Mu Us sandland of China (Yang, 2010). The precipitation threshold varied with factors such as the eco-zone, the PFT, the canopy structure, and the branch architecture. A greater precipitation threshold partly explained why the SF% of trees was smaller than that of shrubs (Llorens and Domingo, 2007). Particularly, the precipitation threshold of xerophytic shrub species was as small as 0.3 mm for *T. vulgaris* at the northern Lomo Herrero of Spain (Belmonte and Romero, 1998), but up to 2.7 mm for *A. farnesiana* at Linares of Mexico (Návar and Bryan, 1990). In this study, at least a 0.9-mm rainfall was necessary to initiate stemflow in *C. korshinskii*, which was in the range of 0.4-1.4 mm at the precipitation threshold for *C. korshinskii* (Li et al., 2009; Wang et al., 2014). This result was consistent with the 0.8 mm for *R. offcinalis* at the northern Lomo Herrero of Spain (Belmont and Romero, 1998) and 0.6 mm for *M. squamosa* at Qinghai-Tibet plateau of China (Zhang et al., 2015). Comparatively, *S. psammophila* needed a 2.1-mm precipitation threshold to initiate



stemflow, which was consistent with the 2.2 mm threshold of *S. psammophila* in the Mu Us
desert (Li et al., 2009) and the 1.9 mm threshold for *R. soongorica* at the west of Loess
Plateau (Li et al., 2008) and the 1.8 mm threshold for *A. ordosica* at the Tengger desert of
China (Wang et al., 2014). Generally, for many xerophytic shrub species, the precipitation
threshold usually ranges between 0.4-2.2 mm, which is in accordance with the findings for
stemflow production ($SF_b$, $SF_d$ and SF%) and the production efficiency (SFP and FR), thus
indicating that rains ≤2 mm were particularly significant for the endemic plants in
water-stressed ecosystems.
Scant rainfall was the most prevalent type in arid and semiarid regions. Rains ≤5 mm
accounted for 74.8% of the annual rainfall events and 27.7% of the annual precipitation
amount at the Anjiapo catchment in the western Loess Plateau of China (with a MAP of 420
mm) (Jian et al., 2014). While at Haizetan in the south of Mu Us sandland of China (with a
MAP of 394.7 mm), rains ≤5 mm accounted for 49.0% of all the rainfall events and 13.8% of
the total precipitation amount of rainy season (lasting from May to September) (Yang 2010).
Additionally, rains ≤2.54 mm accounted for 60% of the total rainfall events and 5.4% of the
total precipitation amount at the eastern Edwards Plateau, the central Texas of USA (with a
MAP of 600-900 mm) (Owens et al., 2006). In this study, rains ≤2 mm accounted for 45.7%
of all the rainfall events and 7.2% of the precipitation amount during the 2014 and 2015 rainy
seasons. In general, *C. korshinskii* and *S. psammophila* produced stemflow during 71 (75.5%
of the total rainfall events) and 51 rainfall events (54.3% of the total rainfall events),
respectively. Because the precipitation threshold for *S. psammophila* was 2.1 mm, 20 rainfall
events of 12-mm, which encompassed 21.3% of all rainfall events, did not produce stemflow,
but stemflow production under these water stress conditions was an extra benefit for *C.*
*korshinskii*. Although the total amount was limited, it was of significant importance for the
survival of the xerophytic shrubs, particularly during long intervals with no rainfall.




In addition to the meteorological characteristics, the canopy structure and branch
architecture partly explained the inter-specific differences in the precipitation threshold
(Crockford and Richardson, 2000; Levia and Frost, 2003). A large, tall canopy created a large
rainfall interception area, also known as "canopy exposure" (Iida et al. 2011), particularly
during windy conditions (Van Stan et al, 2011). However, this advantage in stemflow
production might be offset by more consumption for wetting canopy and evaporation before
stemflow is generated in arid and semiarid regions, in which considerable evapotranspiration
potentially occurs. This phenomenon might be responsible for the smaller precipitation
threshold for stemflow production in *C. korshinskii*, which had a canopy height of $2.1 \pm 0.2$
m and a canopy area of $5.14 \pm 0.26$ m$^2$, than *S. psammophila,* which had a canopy height of
$3.5 \pm 0.2$ m and a canopy area of $21.35 \pm 5.21$ m$^2$. Additionally, the canopy structure and
branch architecture also affected the water holding capacity (Herwitz, 1985), the interception
loss (Dunkerley, 2000), and consequently the precipitation threshold for stemflow generation
(Staelens et al., 2008). Nevertheless, the most influential plant traits had not determined yet,
and further stemflow studies was required at the finer leaf scale and temporal scale in the
future (Levia and Germer, 2015).

**4.3 Secure stemflow production advantage via beneficial leaf traits**
Further studies at the leaf scale indicated that leaf traits had a significant influence on
stemflow (Návar and Bryan, 1990; Carlyle-Mose, 2004; Garcia-Estringana et al., 2010). At
the individual shrub scale, the canopy gap, as represented by the LAI and the leaf mass,
provided direct access for raindrops to the branch surface (Crockford and Richardson, 2000).
The positive effects of LAI (Liang et al., 2009) and leaf biomass (Yuan et al., 2016) have
already been confirmed for *Stewartia monadelpha* and *S. psammophila*, respectively. In a
study of European beech saplings, Levia et al. (2015) assumed that a threshold number of





leaves might exist for stemflow production. The positive effects could become negative if too
many leaves enclose the branches, which would benefit throughfall instead. In general,
factors such as a relatively large number of leaves (Li and Xiao, 2016), a large leaf area (Li et
al., 2015), a scale-like leaf arrangement (Owens et al., 2006), a small individual leaf area
(Sellin et al., 2012) , a concave leaf shape (Xu et al., 2005), a densely veined leaf structure,
an upward leaf orientation (Crockford and Richardson, 2000), leaf pubescence
(Garcia-Estringana et al., 2010), and the leaf epidermis microrelief (e.g., the non-hydrophobic
leaf surface and the grooves within it) (Roth-Nebelsick et al., 2012) together result in the
retention of a large amount of precipitation in the canopy, supplying water for stemflow
production, and providing a beneficial morphology that enables the leaves to function as a
highly efficient natural water collecting and channelling system.

According to the field observations in this study, *C. korshinskii* had better leaf

morphology for stemflow production than did *S. psammophila*, owing to a lanceolate and
concaved leaf shape, a pinnate compound leaf arrangement and a densely sericeous pressed
pubescence (Fig. 2). Additionally, experimental measurements indicated that *C. korshinskii*
had a larger MTA, LAB, LNB and SLW (an average of 54.4 ؛, 2509.05 cm$^2$, 12479 and 126.04
g cm$^{-2}$, respectively) and a smaller ILAB (an average of 21.94 mm$^2$) than did *S. psammophila*
(an average of 48.5 ؛, 1797.93 cm$^2$, 2404, 73.87 g cm$^{-2}$ and 87.52 mm$^2$, respectively). The
larger SLW indicated that more biomass was deposited per unit leaf area. The concave leaf
shape, upward leaf orientation (MTA) and densely veined leaf structure (ILAB) (Xu et al.,
2005) provided stronger leaf structural support in *C. korshinskii* for the interception and
transportation of precipitation, particularly during highly intense rains. Therefore, in addition
to the leaf morphology, *C. korshinskii* was also equipped with more beneficial leaf structural
characteristics for stemflow production.

However, given that BML had strong effects on stemflow in *S. psammophila* (Yuan et al.,




2016), why were stem traits identified as the single most influential traits for stemflow production in *C. korshinskii*, as indicated by the BMS in this study? The answer may partly lie in the values of HV and PBMS. HV was computed as the cross-sectional area of the xylem divided by the total leaf area supported by the stems (Sellin et al., 2012). A higher HV indicates a potentially better water supply to leaves in terms of hydraulic conductance. However, it could also be interpreted as indicating that more stem tissues are required to support the unit leaf area for the normal function of the individual branch. The average HV of *C. korshinskii* was 0.0507 and increased from 0.0438 for the 5–10-mm branches to 0.0615 for the >18-mm branches and was an order of magnitude higher than in *S. psammophila*, which averaged 0.0009 and remained nearly the same for different BD categories. The optimal partitioning theory indicates that plants preferentially allocate biomass into the organs that harvest the most limiting resource (Thornley, 1972; Bloom et al., 1985) and finally reach the "functional equilibrium" of biomass allocation (Brouwer, 1963; Iwasa and Roughgarden, 1984). Therefore, a greater stem biomass might be required by *C. korshinskii* to support leaf development than in *S. psammophila*, thus allowing more carbohydrate produced and raindrops intercepted at the canopy. This possibility is consistent with the biomass allocation patterns and leaf areas of the shrub species in this study. *C. korshinskii* allocated more biomass into the stems with an average of PBMS of 85.6% and had a larger leaf area with an average of LAB of 2509.1 cm$^2$ than *S. psammophila*, which had an average PBMS and LAB of 81.9% and 1797.9 cm$^2$, respectively. The larger values of PBMS and LAB in *C. korshinskii* were observed for all BD categories (Table 1). Additionally, the larger PBMS helped to prevent the intercepted rain drops from falling off under windy conditions, which also benefited stemflow production in *C. korshinskii*.

**5 Conclusions**



Compared with *S. psammophila*, *C. korshinskii* produced a larger amount of stemflow
more efficiently; an average 1.9, 1.3, 1.4, 1.6 and 2.5-fold increase in *C. korshinskii* was
observed for the branch stemflow production ($SF_b$), the shrub stemflow depth ($SF_d$), the shrub
stemflow percentage (SF%), the stemflow productivity (SFP) and the stemflow funnelling
ratio (FR), respectively. The largest specific difference in stemflow production ($SF_b$, $SF_d$ and
SF%) and the production efficiency (SFP and FR) was during rains ≤2 mm, which were the
most frequent rainfall events. Although the total amount of rainfall was limited, it was of
great importance for *C. korshinskii* to survive and thrive, particularly during the extreme
drought period. Additionally, the inter-specific differences in $SF_b$, $SF_d$, SF% and SFP were
maximized for the 5–10-mm branches; this result was particularly significant because it
encouraged young shoots by supplying more water.
Beneficial leaf traits, including a lanceolate and concaved leaf shape, a pinnate
compound leaf arrangement, a densely sericeous pressed pubescence, an upward leaf
orientation (MTA), a large leaf area (LAB), a relatively large number of leaves (LNB), a large
leaf area index (LAI), a small individual leaf area (ILAB), and a large specific leaf weight
(SLW), might be responsible for the superior stemflow production in *C. korshinskii*. Along
with the canopy structure, these leaf traits may account for the lower precipitation threshold
to initiate stemflow in *C. korshinskii* (0.9 mm) than in *S. psammophila* (2.1 mm). A lower
precipitation threshold enabled *C. korshinskii* to harvest more water from rainfall via
stemflow.
In conclusion, a higher and more efficient stemflow, a lower precipitation threshold and
beneficial leaf traits provided *C. korshinskii* with greater drought tolerance and a competitive
edge in a water-stressed ecosystem.

*Acknowledgments*. This research was funded in part by the National Natural Science





Foundation of China (No. 41390462), the National Key Research and Development Program
(No. 2016YFC0501602) and the Youth Innovation Promotion Association CAS (No.
2016040). We are grateful to Mengyu Wang, Dongyang Zhao, Meixia Mi and Hongmin Hao
for field assistant. Special thanks were given to the Shenmu Erosion and Environment
Research Station for experiment support to this research.

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





**Table captions**

**Table 1.** Comparison of leaf traits, branch morphology and biomass indicators of *C.*
*korshinskii* and *S. psammophila*.

**Table 2.** Comparison of stemflow production ($SF_b$, $SF_d$ and $SF\%$) between *C. korshinskii* and
*S. psammophila*.

**Table 3.** Comparison of stemflow productivity (SFP) between *C. korshinskii* and *S.*
*psammophila*.

**Table 4.** Comparison of the funneling ratio (FR) for *C. korshinskii* and *S. psammophila*.





**Table 1.** Comparison of leaf traits, branch morphology and biomass indicators of *C. korshinskii* and *S. psammophila*.

| Plant traits | | *C. korshinskii* (categorized by BD, mm) | | | | | *S. psammophila* (categorized by BD, mm) | | | | |
|---|---|---|---|---|---|---|---|---|---|---|---|
| | | 5-10 | 10-15 | 15-18 | >18 | Avg. (BD) | 5-10 | 10-15 | 15-18 | >18 | Avg. (BD) |
| Leaf traits | LAB (cm$^2$) | 1202.7 | 2394.5 | 3791.2 | 5195.2 | 2509.1 ±1355.3 | 499.2 | 1317.7 | 2515.2 | 3533.6 | 1797.9 ±1118.0 |
| | LNB | 4787 | 11326 | 20071 | 29802 | 12479 ±8409 | 392 | 1456 | 3478 | 5551 | 2404 ±1922 |
| | ILAB (mm$^2$) | 25.4 | 21.3 | 18.9 | 17.5 | 21.9 ±3.0 | 135.1 | 93.1 | 72.6 | 64.3 | 93.1 ±27.8 |
| | SLW (g cm$^2$) | 126.4 | 126.0 | 125.7 | 125.6 | 126.0 ±0.3 | 95.6 | 74.5 | 63.0 | 58.1 | 73.9 ±14.5 |
| | HV | 0.0438 | 0.0513 | 0.0572 | 0.0615 | 0.0507 ±0.0064 | 0.0010 | 0.0009 | 0.0009 | 0.0009 | 0.0009 ±0.0001 |
| Branch morphology | BD (mm) | 8.17 | 12.49 | 16.61 | 20.16 | 12.48 ±4.16 | 7.91 | 12.48 | 16.92 | 19.76 | 13.73 ±4.36 |
| | BL (cm) | 137.9 | 160.3 | 195.9 | 200.7 | 161.5 ±35.0 | 212.5 | 260.2 | 290.4 | 320.1 | 267.3 ±49.7 |
| | BA (°) | 63 | 56 | 63 | 64 | 60 ±18 | 64 | 63 | 51 | 60 | 60 ±20 |
| Biomass indicators | BML (g) | 13.9 | 19.0 | 30.2 | 41.4 | 19.9 ±10.8 | 5.4 | 18.0 | 40.0 | 61.3 | 27.9 ±20.7 |
| | BMS (g) | 62.9 | 121.4 | 236.4 | 375.8 | 141.1 ±110.8 | 23.0 | 81.4 | 188.5 | 295.5 | 130.7 ±101.4 |
| | PBMS (%) | 82.0 | 86.3 | 88.7 | 90.0 | 85.6 ±3.1 | 80.8 | 81.8 | 82.5 | 82.8 | 81.9 ±0.8 |

Note: LAB and LNB are leaf area and number of branch, respectively. ILAB is individual leaf area of branch. SLW is the specific leaf
weight, and HV was the Huber value. BD, BL and BA are average branch basal diameter, length and angle, respectively. BML and BMS
are biomass of leaves and stems, respectively. PBMS is the percentage of leaf biomass to that of branch. The average values mentioned
above are expressed as the means ± SE.



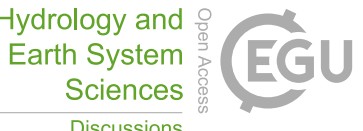

**Table 2.** Comparison of stemflow production ($SF_b$, $SF_d$ and $SF\%$) between *C. korshinskii* and *S. psammophila*.

| Intra- and inter-specific differences | Stemflow indicators | BD categories (mm) | Precipitation categories (mm) | | | | | | Avg.(P) |
|---|---|---|---|---|---|---|---|---|---|
| | | | ≤2 | 2-5 | 5-10 | 10-15 | 15-20 | >20 | |
| Intra-specific differences in *C. korshinskii* (*CK*) | $SF_b$ (mL) | 5-10 | 10.7 | 29.8 | 73.5 | 109.9 | 227.6 | 306.1 | 119.0 |
| | | 10-15 | 26.0 | 64.0 | 166.1 | 236.0 | 478.6 | 689.7 | 262.4 |
| | | 15-18 | 44.3 | 103.3 | 279.9 | 416.6 | 826.0 | 1272.3 | 464.5 |
| | | >18 | 69.5 | 145.4 | 424.4 | 631.4 | 1226.9 | 1811.7 | 679.9 |
| | | Avg.(BD) | 28.4 | 67.3 | 180.6 | 264.6 | 529.2 | 771.4 | 290.6 |
| | $SF_d$ (mm) | N/A | 0.09 | 0.24 | 0.63 | 0.91 | 1.85 | 2.64 | 1.00 |
| | $SF\%$ (%) | N/A | 5.8 | 6.6 | 8.8 | 7.5 | 10.1 | 8.9 | 8.0 |
| Intra-specific differences in *S. psammophila* (*SP*) | $SF_b$ (mL) | 5-10 | 2.8 | 8.9 | 28.8 | 47.2 | 66.5 | 120.0 | 43.0 |
| | | 10-15 | 7.6 | 23.2 | 76.6 | 134.6 | 188.3 | 353.5 | 121.8 |
| | | 15-18 | 12.0 | 35.9 | 121.6 | 223.4 | 319.4 | 592.6 | 201.5 |
| | | >18 | 16.2 | 52.3 | 165.5 | 289.2 | 439.6 | 860.4 | 281.8 |
| | | Avg.(BD) | 9.0 | 28.0 | 91.6 | 162.2 | 234.8 | 444.3 | 150.3 |
| | $SF_d$ (mm) | N/A | 0.01 | 0.11 | 0.48 | 0.89 | 1.27 | 2.23 | 0.78 |
| | $SF\%$ (%) | N/A | 0.7 | 3.0 | 6.1 | 6.8 | 7.2 | 7.9 | 5.5 |
| Inter-specific differences (the ratio of the stemflow production of *CK* to that of *SP*) | $SF_b$ | 5-10 | 3.8 | 3.3 | 2.6 | 2.3 | 3.4 | 2.6 | 2.8 |
| | | 10-15 | 3.4 | 2.8 | 2.2 | 1.8 | 2.5 | 2.0 | 2.2 |
| | | 15-18 | 3.7 | 2.9 | 2.3 | 1.9 | 2.6 | 2.2 | 2.3 |
| | | >18 | 4.3 | 2.8 | 2.6 | 2.2 | 2.8 | 2.1 | 2.4 |
| | | Avg.(BD) | 3.2 | 2.4 | 2.0 | 1.6 | 2.3 | 1.7 | 1.9 |
| | $SF_d$ | N/A | 8.5 | 2.2 | 1.3 | 1.0 | 1.5 | 1.2 | 1.3 |
| | $SF\%$ | N/A | 8.3 | 2.2 | 1.4 | 1.1 | 1.4 | 1.1 | 1.4 |

Note: BD is the branch basal diameter; P is the precipitation amount; *CK* and *SP* are the abbreviations of *C. korshinskii* and *S. psammophila*, respectively.



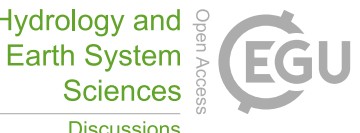

**Table 3.** Comparison of stemflow productivity (SFP) between *C. korshinskii* and *S. psammophila*.

| Intra- and inter-specific differences | BD categories (mm) | Precipitation categories (mm) | | | | | | Avg.(P) |
|---|---|---|---|---|---|---|---|---|
| | | ≤2 | 2-5 | 5-10 | 10-15 | 15-20 | >20 | |
| Intra-specific differences in *C. korshinskii* (*CK*) (mL g⁻¹) | 5-10 | 0.20 | 0.56 | 1.37 | 2.04 | 4.18 | 5.60 | 2.19 |
| | 10-15 | 0.19 | 0.47 | 1.20 | 1.72 | 3.47 | 4.96 | 1.90 |
| | 15-18 | 0.17 | 0.38 | 1.05 | 1.55 | 3.08 | 4.74 | 1.73 |
| | >18 | 0.15 | 0.35 | 1.00 | 1.46 | 2.95 | 4.35 | 1.62 |
| | Avg.(BD) | 0.19 | 0.47 | 1.21 | 1.78 | 3.60 | 5.08 | 1.95 |
| Intra-specific differences in *S. psammophila* (*SP*) (mL g⁻¹) | 5-10 | 0.11 | 0.34 | 1.10 | 1.83 | 2.51 | 4.59 | 1.64 |
| | 10-15 | 0.08 | 0.25 | 0.82 | 1.43 | 1.98 | 3.72 | 1.29 |
| | 15-18 | 0.05 | 0.16 | 0.53 | 0.97 | 1.40 | 2.61 | 0.88 |
| | >18 | 0.05 | 0.15 | 0.47 | 0.82 | 1.25 | 2.44 | 0.80 |
| | Avg.(BD) | 0.07 | 0.23 | 0.76 | 1.31 | 1.84 | 3.43 | 1.19 |
| Inter-specific differences (the ratio of the SFP values of *CK* to that of *SP*) | 5-10 | 1.8 | 1.7 | 1.3 | 1.1 | 1.7 | 1.2 | 1.3 |
| | 10-15 | 2.4 | 1.9 | 1.5 | 1.2 | 1.8 | 1.3 | 1.5 |
| | 15-18 | 2.8 | 2.4 | 2.0 | 1.6 | 2.2 | 1.8 | 2.0 |
| | >18 | 3.0 | 2.3 | 2.1 | 1.8 | 2.4 | 1.8 | 2.0 |
| | Avg.(BD) | 2.7 | 2.0 | 1.6 | 1.4 | 2.0 | 1.5 | 1.6 |

Note: BD is the branch basal diameter; P is the precipitation amount; *CK* and *SP* are the abbreviations of *C. korshinskii* and *S.*
*psammophila*, respectively.




**Table 4.** Comparison of the funneling ratio (FR) for *C. korshinskii* and *S. psammophila*.

| Intra- and inter-specific differences | BA categories (°) | Precipitation categories (mm) | | | | | | Avg.(P) |
|---|---|---|---|---|---|---|---|---|
| | | ≤2 | 2-5 | 5-10 | 10-15 | 15-20 | >20 | |
| Intra-specific differences in *C. korshinskii* (*CK*) | ≤30 | 100.18 | 127.68 | 168.14 | 125.30 | 193.06 | 170.31 | 149.90 |
| | 30-60 | 125.89 | 133.77 | 178.5 | 157.84 | 205.19 | 182.07 | 164.65 |
| | 60-80 | 135.51 | 148.94 | 192.45 | 165.83 | 217.03 | 188.64 | 176.06 |
| | >80 | 133.17 | 167.44 | 205.53 | 182.61 | 276.02 | 226.08 | 198.16 |
| | Avg.(BA) | 129.17 | 144.84 | 187.74 | 162.34 | 219.61 | 190.34 | 173.34 |
| Intra-specific differences in *S. psammophila* (*SP*) | ≤30 | 32.60 | 37.33 | 52.02 | 59.00 | 65.75 | 85.19 | 54.97 |
| | 30-60 | 34.50 | 43.44 | 65.67 | 70.63 | 77.74 | 92.28 | 64.78 |
| | 60-80 | 37.83 | 47.92 | 77.99 | 78.41 | 82.31 | 97.72 | 72.39 |
| | >80 | 44.88 | 54.99 | 93.45 | 94.74 | 94.09 | 115.72 | 85.57 |
| | Avg.(BA) | 36.65 | 46.01 | 72.57 | 75.34 | 80.45 | 96.09 | 69.25 |
| Inter-specific differences (the ratio of the FR values of *CK* to that of *SP*) | ≤30 | 3.1 | 3.4 | 3.2 | 2.1 | 2.9 | 2.0 | 2.7 |
| | 30-60 | 3.7 | 3.1 | 2.7 | 2.2 | 2.6 | 2.0 | 2.5 |
| | 60-80 | 3.6 | 3.1 | 2.5 | 2.1 | 2.6 | 1.9 | 2.4 |
| | >80 | 3.0 | 3.0 | 2.2 | 1.9 | 2.9 | 2.0 | 2.3 |
| | Avg.(BA) | 3.5 | 3.2 | 2.6 | 2.2 | 2.7 | 2.0 | 2.5 |


Note: BA is the branch inclined angle; P is the precipitation amount; *CK* and *SP* are the abbreviations of *C. korshinskii* and *S. psammophila*, respectively.



**Figure captions**

**Fig. 1.** Location of the experimental stands and facilities for stemflow measurements of *C. korshinskii* and *S. psammophila* at the Liudaogou catchment in the Loess Plateau of China.

**Fig. 2.** Comparison of leaf morphologies of *C. korshinskii* and *S. psammophila*.

**Fig. 3.** Verification of the allometric models for estimating the biomass and leaf traits of *C. korshinskii*. BML and BMS refer to the biomass of the leaves and stems, respectively, and LAB and LNB refer to the leaf area and the number of branches, respectively.

**Fig. 4.** Relationships of branch stemflow production ($SF_b$), shrub stemflow depth ($SF_d$) and stemflow percentage ($SF\%$) with precipitation amount (P) for *C. korshinskii* and *S. psammophila*.





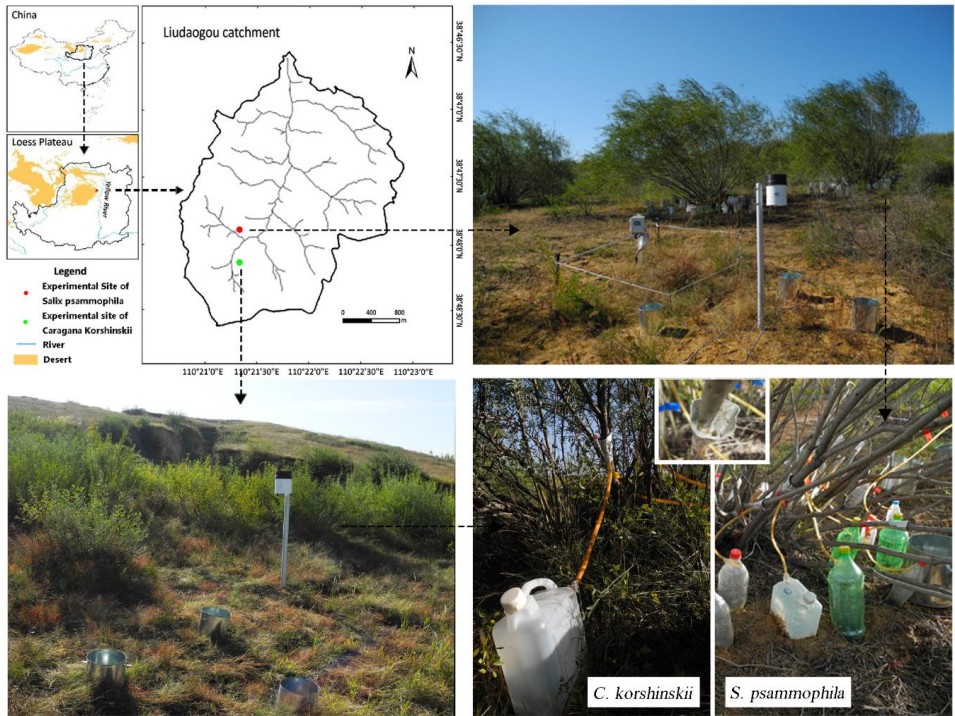

**Fig. 1.** Location of the experimental stands and facilities for stemflow measurements of *C. korshinskii* and *S. psammophila* at the Liudaogou catchment in the Loess Plateau of China.





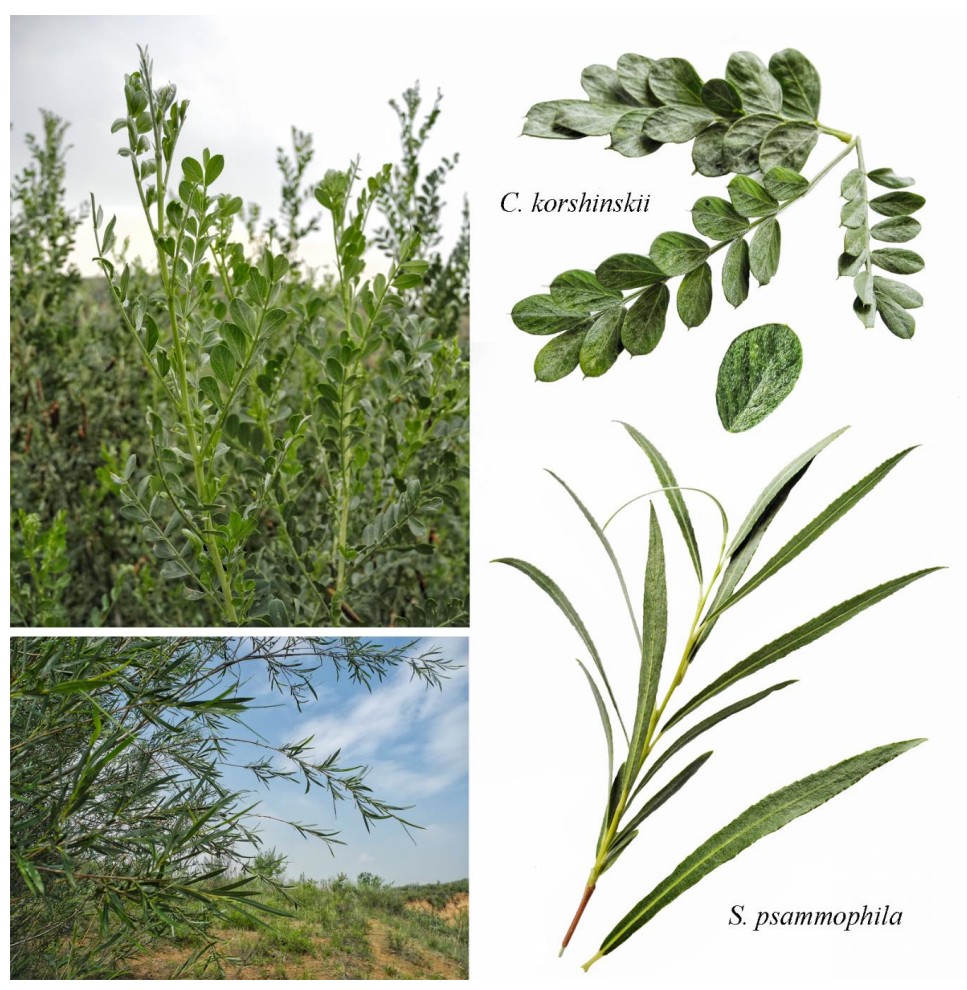


**Fig. 2.** Comparison of leaf morphologies of *C. korshinskii* and *S. psammophila*.





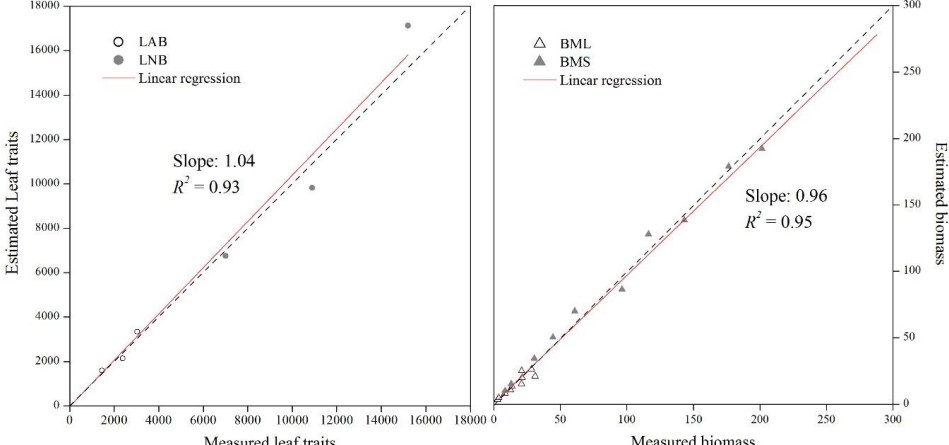


**Fig. 3.** Verification of the allometric models for estimating the biomass and leaf traits of *C. korshinskii*. BML and BMS refer to the biomass of the leaves and stems, respectively, and LAB and LNB refer to the leaf area and the number of branches, respectively.





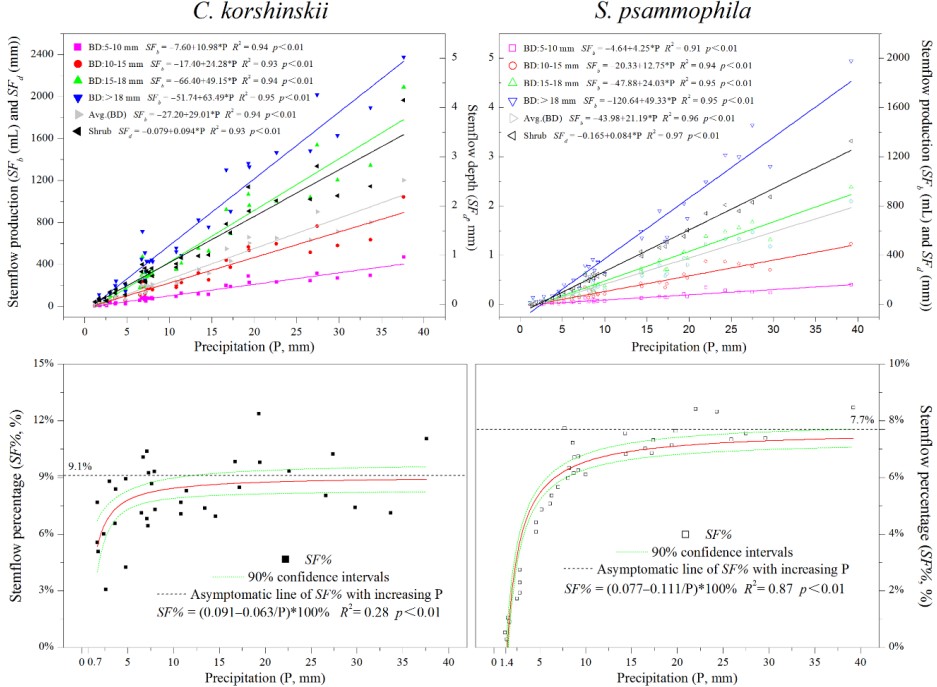


**Fig. 4.** Relationships of branch stemflow production ($SF_b$), shrub stemflow depth ($SF_d$) and
stemflow percentage ($SF\%$) with precipitation amount (P) for *C. korshinskii* and *S. psammophila*.