# Peer review of "Comparisons of stemflow and its bio-/abiotic influential factors"

_Hydrology and Earth System Sciences, 2016_

## Referee Comment (RC1) · D. Dunkerley (Referee) · 9 Sep 2016

This paper reports field data on stemflow volumes from a dryland field site in China, collected over two successive annual wet seasons. The paper is systematically presented, though rather too long in light of the scope and volume of the primary data that are presented. The field data are of interest because they include stemflow measurements at the scale of individual branches.

I felt that the authors needed some evidence to support their repeated claims (e.g. line 58-59) that stemflow exerts a high influence on the survival of dryland shrubs, especially under drought conditions (e.g. line 107 refers to ' . . . a novel characterization of plant drought tolerance. . .' as one of the outcomes proposed for the present study).

[Figure]

The authors collected only data on rainfall and on stemflow volumes. They did not record soil moisture near the plant stems, or observe the fate of stemflow near the soil surface – where, for instance, it might be involved in lateral flow through organic litter materials, or indeed trickle away as overland flow. Instead, they were content to assume tacitly that all of the stemflow was plant-available. Soils are only briefly described, but the authors do note in passing that the surface textures differed between the two shrub species examined (refer to lines 136-137), one being loess and the other, sand.

Field experiments were conducted only during the rainy season (line 143) but about a quarter of the annual rainfall comes in the drier season, and I think that conditions then needed to be considered also, as the longer, 8 month dry season is possibly the time when plant available moisture is more critical.

Only four individuals of each species were instrumented to collect stemflow data. This is not a large sample, though I appreciate the tedium of instrumenting multi-stemmed plants. Furthermore, of the four plants, only about one third of the branches were instrumented for C. korshinskii, and less than half for S. psammophila. This reduces the effective sample size still further.

Given that it has often been reported that stemflow may fall from branches when rain becomes intense (and overtaxes the ability of stems to conduct all of the incident water), I wondered about the possible effects of trapping and diverting stemflow from so many branches into collecting vessels. This presumably reduced branch drip and so, perhaps, the branch flow carried by branches lying beneath higher ones from which the stemflow had been diverted. I think that the authors need to consider and discuss this possibility, in relation to the possible path of rainfall and throughfall (both free and released) through the canopy of these shrubs.

Relevant field data that I would have liked to see included in the paper are on air temperature, humidity, and windspeed. Solar radiation data would also be informative,
together with data on whether the rainfall was recorded primarily during daylight hours or at night, since this is relevant to evaporative losses and to the efficiency with which stemflow can be conveyed across the plant surfaces. The authors can hopefully shed light on at least some of these issues.

The authors are imprecise when reporting their results. For instance, line 287 reports average branch stemflow volumes in mL, but the authors do not state whether this is across all rainfall, or averaged per rainfall event, or processed in some other way. For reported stemflow volumes, the associated time period must be stated. Likewise, in line 297, 298, etc., are the volumes reported the sum of stemflow for all branches or the mean per branch or something else? The reporting needs to be much clearer. It is the same when the authors discuss funneling ratios in line 342 and following. Are the figures in this section ratios for individual rainfall events, or averaged over all events? As mentioned earlier, the authors also need to consider how the complete trapping of stemflow from upper branches might have affected the stemflow on lower branches, that might have received less drip from above.

I felt that the authors were vague in their discussion of other results. For instance, lines 366-367 state that precipitation amount was the most important rainfall characteristic that affected stemflow in the studied shrub species. Here I presume they mean that precipitation amount had affected aggregate stemflow volume (and presumably measured at rainfall event scale). Other aspects of stemflow, for instance the peak flux or rate of delivery of stemflow to the base of the plant, are much more likely to have been affected by rainfall intensity. I am not sure why the authors only consider overall stemflow volume, and they should make a case for neglecting other ways to characterize stemflow, including the timing of its delivery from the plant. Stemflow volume alone does not provide a complete exploration of the origin and fate of stemflow.

The fundamental argument of the paper is again in need of supporting evidence from the beginning of the Discussion at line 393. The authors discuss 'effective utilization' of precipitation but as pointed out above, have no data relating to this. Their data only

estimate stemflow volumes on above-ground parts of the plants. How this translates to soil moisture in the root zone (allowing for evaporation and interception on litter) is not clear. The authors should not make claims that are not supported (or supportable) using their available data. They argue in lines 404-405 about the 'effective utilization of precipitation' by the two shrub species in rainfalls of < 2 mm. However, any stemflow delivered to the base of the shrubs in what are likely to be short showers, might be largely lost to evaporation once the short event ended. This should illustrate how spurious it might be to infer utilization from stemflow data not supported by soil moisture data, or indeed by measures of transpiration by the plants. The authors proceed (e.g. line 420) to argue about energy conservation, again speculating about the utilization of stemflow from rainfall events of < 2 mm. All of this is completely unsupported by the data, and should be eliminated from the paper, or at least highlighted as completely speculative. Again, in line 430-431 the authors speculate about drought tolerance; not only do they have no supporting data, but the data that they do have were derived during the rainy season, and not in drought conditions at all. How the shrub foliage etc. might change during drought years remains unknown and the authors should eliminate all of their speculation about drought tolerance. Their data relate to stemflow alone, and they should restrict themselves primarily to discussing and interpreting those data. Lines such as 476-478 inclusive are completely speculative, though the authors write as though they are presenting a result from their work. They refer to stemflow production under 'water stress conditions' though they did not observe this; they refer to their estimated stemflow being 'of significant importance for the survival of the xerophytic shrubs, particularly during long intervals with no rainfall' though they present absolutely no evidence to support this claim, having no data from long periods with no rainfall. All of this speculation should be eliminated from the paper, or at the very least identified as speculation not supported by any data. Overall, the focus of the paper needs to shift from speculation to the discussion of what can validly be determined from the field evidence available, namely, the estimated stemflow volumes.

Minor errors:

Line 41: what are 'stemflow channels'? Does this imply fixed pathways? Line 41: 'pointedly' should be 'directly' or similar. Line 44: what is meant by 'biogeochemical reactivity at the terrestrial-aquatic interface'? Line 58: please cite references to support the claim about 'disproportionately high influence [of stemflow] on survival and competitiveness of xerophytic shrub species'. Line 81: insert missing space before 'Murakami' Line 155: how do branches exist ' as independent individuals'? Line 214: 'at the' should be 'in a' Line 238: should '4080-mm' be '40-80 mm'? Line 268 and many other instances: do not write '18-mm'; the hyphen is not allowed in the SI metric system. There must be a space between the numerical quantity and the symbol for the unit of measurement (e.g. '18 mm' is correct). Line 280: do the authors data justify 4 decimal places of precision? This requires fixing in many places, such as line 475. Line 475: should 'events of 12-mm' read 'events of 1-2 mm'? Line 492: 'had not determined yet' should read 'have not yet been determined'.

David Dunkerley Monash University

---

## Referee Comment (RC2) · Anonymous Referee #2 · 22 Sep 2016

General comments: This study explored stemflow yield in relations to rainfall characteristics and the plant traits of branches and leaves for two dominant shrubs (C. korshinskii and S. psammophila) during rainy seasons in the northern Loess Plateau of China. This manuscript reports important data on stemflow measurements at the scale of individual branches and highlights the effect of canopy structure (e.g. biomass, the leaf area of the branches, the leaf numbers of the branches, stemflow productivity, and the funnelling ratio) on stemflow production. The finding of this study is interesting and fall into the scope of the HESS. However, my main concern is the title, results and discussions are not really robust and can't be fully supported by data, and the interpretation is weak. The specific comments are listed as follows: (1) Title: The "the effects of leaves

and implications in drought tolerance" in the title is not well reflected in the results of this study. Although measurements of leaf area index (LAI), the foliage orientation, the leaf area of the branches and the leaf numbers of the branches were made in the study, results of species-specific variation of plant traits (line 236-283) just mainly qualitatively described leaf traits, branch morphology and biomass, which were not directly linked with stemflow characteristics. Moreover, results of this study indicated that precipitation amount was the most influential rainfall characteristic and stem biomass and leaf biomass were the most influential plant traits that affected stemflow in C. korshinskii and S. psammophila, so the effects of leaves on stemflow were not well investigated in this study. In the case of implications in drought tolerance, authors mainly discussed with personal speculations, there were not solid soil water data to verify it. So I suggest author could delete "the effects of leaves and implications in drought tolerance" from the title. (2) Introduction: The objectives of this study were not clear, what's the new findings made by this study? What's the knowledge gaps in stemflow researches for shrubs? In fact, stemflow of C. korshinskii and S. psammophila were already studied in China, what's the difference between studies? I wonder if authors can highlight the stemflow yield from branches and stemflow productivity between shrubs. (3) Materials and Methods: As shrubs grow during the rainy period, at what period (time) or measurement frequency do authors measure plant traits, particularly for biomass (line 175), how can you confirm them represent real plant trait dynamics, which were not clearly described in the text. Line 155: what's the "modular organisms and multi-stemmed shrub"? (4) Results: For the most part of the "3.1 Species-specific variation of plant traits", it is not really the results of the study, I would suggest authors move some of the description of C. korshinskii and S. psammophila to the section of "Materials and Methods". Line 387-390: it is not clear, why big difference existed between rains $\leq 10$ mm and the heavy rain. (5) Discussions: I would suggest authors focus on the interpretation of the results of this study, but not speculations on utilization of more rains via a low precipitation, there was not direct evidence or robust data to support the proposed conclusion. (6) English languages needs refine by a native English speakers.

[Figure]

---

## Author Comment (AC1) · 23 Nov 2016

Dear Prof. Wang,

We have substantially revised our manuscript entitled as "Comparisons of stemflow yield and efficiency between two xerophytic shrubs: the effects of leaves and implications in drought tolerance" after considering all the comments made by Prof. David Dunkerley and another anonymous reviewer. These comments were of great help to improve the overall quality of this manuscript.

Please see the attached supplement (.pdf) for the detailed replies to all the comments of Reviewer #1 and Reviewer #2.

Please also note the supplement to this comment:
http://www.hydrol-earth-syst-sci-discuss.net/hess-2016-420/hess-2016-420-AC1-supplement.pdf

─────────────────────────

**Fig. 1.** Location of the experimental stands and facilities for stemflow measurements of C. korshinskii and S. psammophila at the Liudaogou catchment in the Loess Plateau of China.

Foliated

Defoliated

*C. korshinskii*

*S. psammophila*

**Fig. 2.** The controlled experiment for stemflow yield between the foliated and manually defoliated shrubs.

[Figure]

**Fig. 3.** Meteorological characteristics of rainfall events for stemflow measurements during the 2014 and 2015 rainy seasons.

[Figure]

**Fig. 4.** Verification of the allometric models for estimating the biomass and leaf traits of C. korshinskii.

**C. korshinskii**

**S. psammophila**

**Fig. 5.** Relationships of branch stemflow volume (SFb), shrub stemflow depth (SFd) and stem-flow percentage (SF%) with precipitation amount (P) for C. korshinskii and S. psammophila.

*C. korshinskii*

*S. psammophila*

**Fig. 6.** Comparison of leaf morphologies of C. korshinskii and S. psammophila.

**Supplement:**

**Revision Notes**

[Figure]

**中国科学院生态环境研究中心**
**Research Center for Eco-Environmental Science**
**Chinese Academy of Sciences (CAS)**
北京市海淀区双清路 18 号 邮编：100085  Website: www.rcees.ac.cn
**18 Shuangqing Road, Haidian District, Beijing 100085, P. R. China**
Tel: 0086-10-62911239  Email: gygao@rcees.ac.cn

November 23, 2016

Memorandum

To:   Prof. Lixin Wang, Editor of *Hydrology and Earth System Science*

Subject: **Revision of hess-2016-420**

Dear Prof. Wang,

  We have substantially revised our manuscript entitled as "Comparisons of stemflow yield and efficiency between two xerophytic shrubs: the effects of leaves and implications in drought tolerance" after considering all the comments made by Prof. David Dunkerley and another anonymous reviewer. These comments were of great help to improve the overall quality of this manuscript.

  The following are the general reply and point-to-point response to all the comments, including (1) Response to Reviewer #1 (Prof. David Dunkerley), (2) Response to Reviewer #2, (3) Revised manuscript with changes marked, and (4) the revised manuscript with no changes marked, respectively.

[Figure]

**中国科学院生态环境研究中心**
**Research Center for Eco-Environmental Science**
**Chinese Academy of Sciences (CAS)**
北京市海淀区双清路 18 号 邮编：100085  Website: www.rcees.ac.cn
**18 Shuangqing Road, Haidian District, Beijing 100085, P. R. China**
Tel: 0086-10-62911239   Email: gygao@rcees.ac.cn

**Response to Reviewer #1, Prof. David Dunkerley:**

**General reply:**

**R1C1:** This paper reports field data on stemflow volumes from a dryland field site in China, collected over two successive annual wet seasons. The paper is systematically presented, though rather too long in light of the scope and volume of the primary data that are presented. The field data are of interest because they include stemflow measurements at the scale of individual branches.

**Reply:**

Thank you for your constructive advices and the "minor revision" recommendation for this manuscript, which has been revised from the following aspects.

1) Some speculative discussion has been deleted in the revised version, and the focus of this work has been shifted to interpret and discuss the measured stemflow data (see Reply to R1C10, please).

2) To explain leaf's effects affecting stemflow yield, a direct evidence has been provided with a controlled experiment of comparing stemflow yield between the foliated and manually defoliated shrubs during the 2015 rainy season (in P.11, Line 235–251, in P.18, Line 433–447, in P.31, Line 758–778, in P.33, Line 812–820 and P.50, Line 1107–1110).

3) To demonstrate the effectiveness in analyzing the abiotic influential factors on stemflow yield and efficiency, more critical meteorological characteristics have been added, including the air temperature, air relative humidity, wind speed and solar radiation in P.9, Line 196–201, in P.14, Line 324–334, and from P.21, Line 529–535.

**Reply for comments on Introduction:**

**R1C2:** I felt that the authors needed some evidence to support their repeated claims (e.g. line 58-59) that stemflow exerts a high influence on the survival of dryland shrubs, especially under drought conditions (e.g. line 107 refers to '….a novel characterization of plant drought tolerance….' as one of the outcomes proposed for the present study).

**Reply:**

Thank you for this comment. New references have been cited as required to support the claim that "stemflow exerts a high influence on the survival of dryland shrubs, especially under drought conditions" in P.4, Line 72–78.

Besides, we have deleted the claim for "a novel characterization of plant drought tolerance", and re-addressed the research objectives and outcomes in P.7, Line 139–142: "The achievement of these research objectives would advance our understanding of the ecological importance of stemflow for dryland shrubs and the significance of leaves from an eco-hydrological perspective".

**Reply for comments on experiment design:**

**R1C3:** The authors collected only data on rainfall and on stemflow volumes. They did not record soil moisture near the plant stems, or observe the fate of stemflow near the soil surface – where, for instance, it might be involved in lateral flow through organic litter materials, or indeed trickle away as overland flow. Instead, they were content to assume tacitly that all of the stemflow was plant-available. Soils are only briefly described, but the authors do note in passing that the surface textures differed between the two shrub species examined (refer to lines 136-137), one being loess and the other, sand.

**Reply:**

Thank you for commenting on the experimental design of this study. We did not take soil moisture and the relevant fluxes above or under the ground into account at this manuscript, and the reasons were as follow:

1) The objectives of this study.

We aimed to quantify and compare stemflow yield and efficiency of *C. korshinskii* and *S. psammophila* at branch and shrub scales, to explore the biotic influential mechanism particularly at a finer leaf scale, and to identify the most influential meteorological characteristics. Therefore, only the aboveground eco-hydrological process was involved (from P.6, Line 128 to P.7, Line 139), which was illustrated by the following Fig. R1-1.

2) Different surface soil textures.

As pointed in this comment, the surface soil texture differed between the two experimental stands: sand for *S. psammophila* and loess for *C. korshinskii*, respectively. So, it was difficult to compare the contributions of stemflow to the soil moisture dynamics between those two shrub species.

Therefore, in terms of the specific research objectives and the actual stand conditions, we focused on the inter- and intra-specific difference of stemflow yield and efficiency and its bio-/abiotic influential factors between *C. korshinskii* and *S. psammophila* at this manuscript. But, given that stemflow was well documented as an important source of available moisture at dryland ecosystems (Dunkerley, 2000; Yang, 2010; Navar, 2011; Li, et al., 2013) (in P.24, Line 594–598), it was necessary and of great significance to explore the relation between stemflow and soil moisture dynamics. This has been listed in our following research plans.

[Figure]

Fig. R1-1. The conceptual framework describing the research objectives and scope: stemflow yield and efficiency and its bio-/abiotic influential factors of *C. korshinskii* and *S. psammophila*.

**R1C4:** Field experiments were conducted only during the rainy season (line 143) but about a quarter of the annual rainfall comes in the drier season, and I think that conditions (in drier season) then needed to be considered also, as the longer, 8 month dry season is possibly the time when plant available moisture is more critical.

**Reply:**

Thank you for this advice on continuing experiments in drier season. It is indeed important for the survival of dryland shrubs to receive enough water supply during dry period.

But different from the Mediterranean climate area, the dry season is the cold and dormant season at the experimental sites. During this period, most of dryland shrubs, including *S. psammophila* and *C. korshinskii*, defoliate. Despite of less precipitation supply, there is less water demand as well. On the contrary, the rainy season was the warm and growing season at this area. During this period, the dryland shrubs foliate, bloom, reproduce and compete with each other for lights and water. The greater water demand makes them more sensitive to the precipitation variation. It is common for these dryland shrubs to experience several wetting-drying cycles (Cui and Caldwell, 1997), especially at northern Loess Plateau of China, where rains are sporadic (in P.24, Line 583–594). Therefore, how to employ the precipitation pulse and small rains to improve water availability is of great importance for dryland shrubs at the rainy season. As an important water resource for soil available moisture, to produce stemflow with a great amount in an efficient manner might be an effective strategy to acquire water (Murakami, 2009) and withstand drought (Martinez-Meza and Whitford, 1996) (in P.24, Line 594–598).

Nevertheless, it indeed makes this study more systematical and convincing to involve stemflow measurements in drier season. We would consider it seriously in the future, if condition permits.

**R1C5:** Only four individuals of each species were instrumented to collect stemflow data. This is not a large sample, though I appreciate the tedium of instrumenting multi-stemmed plants. Furthermore, of the four plants, only about one third of the branches were instrumented for *C. korshinskii*, and less than half for *S. psammophila*. This reduces the effective sample size still further.

**Reply:**

Thank you for commenting on the effective sample size of this study.

Prior to explaining the effective sample size, it is necessary to introduce that both of *C. korshinskii* and *S. psammophila* are the modular organisms, whose zygote develops into a discrete unit (module), and then produces more units like itself, rather than developing into a complete organism (Allaby, 2010). Each module seeks its own survival goals and the resulting organism level behavior is not centrally controlled (Firn, 2004) (in P.9, Line 202–205). It is required to involve both of the genets (shrubs) and ramets (branches) while counting the sample size of modular organisms (He, 2004).

The branches of *S. psammophila* and *C. korshinskii* compete with each other for lights and water, which are the ideal experiment objects to study stemflow at the branch scale (in P.9, Line 204–207). Thus, in this study, we experimented on individual branches and ignored the canopy variance by selecting sample shrubs with similar intra-specific canopy area and height, e.g., 2.1 ± 0.2 m and 5.1 ± 0.3 m² for *C. korshinskii*, and 3.5 ± 0.2 m and 21.4 ± 5.2 m² for *S. psammophila*. A total of 53 branches of *C. korshinskii* (17, 21, 7, 8 for the basal diameter categories of 5–10 mm, 10–15 mm, 15–18 mm and >18 mm, respectively) and 98 branches of *S. psammophila* (20, 30, 20 and 28 branches at the BD categories 5–10 mm, 10–15 mm, 15–18 mm and >18 mm, respectively) were selected for stemflow measurements (in P.10, Line 217–220). Although it is not a great sample size in shrubs amount, it might be enough to discuss stemflow yield and efficiency and the influential mechanism at branch scale.

**R1C6:** Given that it has often been reported that stemflow may fall from branches when rain becomes intense (and overtaxes the ability of stems to conduct all of the incident water), I wondered about the possible effects of trapping and diverting stemflow from so many branches into collecting vessels. This presumably reduced branch drip and so, perhaps, the branch flow carried by branches lying beneath higher ones from which the stemflow had been diverted. I think that the authors need to consider and discuss this possibility, in relation to the possible path of rainfall and throughfall (both free and released) through the canopy of these shrubs.

**Reply:**

Thank you for commenting on the possible effects of experimental setting on stemflow measurements.

In this study, we installed one aluminum foil collar to trap stemflow at one branch, which were fitted around the entire branch circumference and close to the branch base. The installed position and the weight of aluminum foil collars ensured limited effects on the original branch inclination. Besides, nearly all sample branches were selected on the skirts of the crown, where was more convenient for installation and ensured the sample branches with limited shading by other branches lying above as well. Associated with the limited external diameter of foil collars, that minimized the accessing of throughfall (both free and released) (in P.10, Line 223–228). Additionally, other selection criteria were also applied: 1) no intercrossing stems, and 2) no turning point in height from branch tip to the base, so as to avoid stemflow converging and bypassing under the influence of neighboring branches and the irrelevant drip-offs (the released throughfall) (Dong, et al., 1987). After completing measurements, the stemflow was returned to the branch base to mitigate the unnecessary drought stress for the sample branches. By doing so, we tried the best to measure the authentic stemflow yield at branch scale with least unnecessary disturbance, including the effects of free and released throughfall on stemflow measurements at this manuscript (from P. 10, Line 230 to P.11, Line 234).

**R1C7:** Relevant field data that I would have liked to see included in the paper are on air temperature, humidity, and windspeed. Solar radiation data would also be informative, together with data on whether the rainfall was recorded primarily during daylight hours or at night, since this is relevant to evaporative losses and to the efficiency with which stemflow can be conveyed across the plant surfaces. The authors can hopefully shed light on at least some of these issues.
**Reply:**

Thank you for commenting on the abiotic influential mechanism of stemflow yield and efficiency. Actually, as shown at the following Fig. R1-2, the meteorological station has been installed to automatically record the wind speed and direction (Model 03002, R. M. Young Company, Traverse City, Michigan, USA), the air temperature and humidity (HMP 155, Vaisala, Helsinki, Finland), and the solar radiation (CNR 4 net radiometer, Kipp & Zonen B.V., Delft, the Netherland). These description has been supplemented in P.9, Line 196–201, and the picture of meteorological station had been updated in Fig.1 in P.55, Line 1142–1144. The detailed meteorological characteristics of rainfall events for stemflow measurements had been supplemented at the "Result" section in P.14, Line 324–334 and indicated by the Fig. 3 in P.58, Line 1149–1151.The relation of meteorological characteristics with stemflow yield and efficiency has been re-analyzed (e.g., indicated at the following Table R1-1 and Table R1-2), and the new findings had been updated from P.20, Line 501 to P.21, Line 506.

[Figure]

Fig. R1-2. The meteorological station was installed to record the wind speed and direction, the air temperature and humidity, and the solar radiation at Liudaogou catchment.

Table R1-1. The significant meteorological characteristics related with the branch stemflow volume ($SF_b$) tested by the Pearson and partial correlation analysises.

| Shrub species | Significant correlation ($p <0.05$) | Non-significant correlation ($p >0.05$) |
|---|---|---|
| *C. korshinskii* | P, $I_{10}$, RD, H | I, $I_5$, $I_{30}$, RI, WS, T, SR |
| *S. psammophila* | P, $I_5$, $I_{10}$, $I_{30}$ | I, RD, RI, WS, T, H, SR |

Note: P means the incident precipitation amount; I, $I_5$, $I_{10}$, $I_{30}$ are the average rainfall intensity, and the maximum rainfall intensity in 5, 10, and 30 minutes, respectively; RD is rainfall duration; RI is rainfall intervals; WS is the wind speed; T and H are the air temperature and humidity, respectively; SR is the solar radiation.

Table R1-2. The relation of branch stemflow volume ($SF_b$) with meteorological characteristics.

| Shrubs | BD categories (mm) | Regression models | $R^2$ | VIF | AIC | Contributions to $SF_b$ (%) P | Contributions to $SF_b$ (%) $I_{10}$ |
|---|---|---|---|---|---|---|---|
| *C. korshinskii* | 5–10 | $SF_b = -7.60+10.98*P$ | 0.94 | 1 | 235.6 | 100 | 0 |
| | | $SF_b = -0.29+11.86*P-1.14*I_{10}$ | 0.96 | 1.2 | 217.4 | 85.7 | 14.3 |
| | 10–15 | $SF_b = -17.40+24.28*P$ | 0.93 | 1 | 296.4 | 100 | 0 |
| | | $SF_b = 2.64+26.94*P-3.36*I_{10}$ | 0.97 | 1.2 | 264.5 | 82.0 | 18.0 |
| | 15–18 | $SF_b = -66.40+49.15*P$ | 0.94 | 1 | 338.9 | 100 | 0 |
| | | $SF_b = -32.91+53.75*P-5.77*I_{10}$ | 0.97 | 1.2 | 313.5 | 84.1 | 15.9 |
| | >18 | $SF_b = -51.74+63.49*P$ | 0.95 | 1 | 348.3 | 100 | 0 |

|  |  |  |  |  |  |  |  |
|---|---|---|---|---|---|---|---|
|  | | $SF_b = -19.50+67.89*P-5.53*I_{10}$ | 0.97 | 1.2 | 333.5 | 87.5 | 12.5 |
|  | Avg.(BD) | $SF_b = -27.20+29.01*P$ | 0.95 | 1 | 298.7 | 100 | 0 |
|  | | $SF_b = -7.46+31.64*P-3.33*I_{10}$ | 0.98 | 1.2 | 271.3 | 84.4 | 15.6 |
| *S. psammophila* | 5–10 | $SF_b = -4.66+21.19*P$ | 0.96 | 1 | N/A | 100 | 0 |
|  | 10–15 | $SF_b = -20.21+12.74*P$ | 0.94 | 1 | N/A | 100 | 0 |
|  | 15–18 | $SF_b = -47.78+24.03*P$ | 0.95 | 1 | N/A | 100 | 0 |
|  | >18 | $SF_b = -120.99+49.35*P$ | 0.96 | 1 | N/A | 100 | 0 |
|  | Avg.(BD) | $SF_b = -43.99+21.19*P$ | 0.96 | 1 | N/A | 100 | 0 |

Note: P is the incident precipitation amount; $I_{10}$ is the maximum rainfall intensity in 10 minutes; BD is the branch basal diameter; VIF is the variance inflation factor; AIC is the Akaike information criterion; $R^2$ is the code of determination; N/A refers to not applicable.

**Reply for comments on Results and Discussion:**

**R1C8:** The authors are imprecise when reporting their results. For instance, line 287 reports average branch stemflow volumes in mL, but the authors do not state whether this is across all rainfall, or averaged per rainfall event, or processed in some other way. For reported stemflow volumes, the associated time period must be stated. Likewise, in line 297, 298, etc., are the volumes reported the sum of stemflow for all branches or the mean per branch or something else? The reporting needs to be much clearer. It is the same when the authors discuss funneling ratios in line 342 and following. Are the figures in this section ratios for individual rainfall events, or averaged over all events? As mentioned earlier, the authors also need to consider how the complete trapping of stemflow from upper branches might have affected the stemflow on lower branches, that might have received less drip from above.

**Reply:**

Thank you for commenting on some imprecise or vague expressions at this manuscript.

We have checked this manuscript carefully and revised these imprecise expressions as required, e.g., adding the corresponding time period in P.17, Line 397, Line 407, Line 414 and Line 419, in P.19. Line 455 and Line 472, in P.23, Line 556 and Line 426, adding the description regarding the sum or the average value for different rainfall events in P.17, Line 397, Line 407 and Line 419, in P.19, Line 454, Line 472 and Line 473–475, in P.21, Line 526, and in P.23, Line 556 and Line 567, and the description regarding the sum or average value for different plant traits in P.17, Line 399 and Line 407, in P.20, Line 476–477, and in P.23, Line 555 and Line 557–558.

The experimental setting for stemflow collection has been explained at Reply for R1C6, in which we described the practices on how to minimize the influences on the authentic branch stemflow measurements.

**R1C9:** I felt that the authors were vague in their discussion of other results. For instance, lines

366-367 state that precipitation amount was the most important rainfall characteristic that affected stemflow in the studied shrub species. Here I presume they mean that precipitation amount had affected aggregate stemflow volume (and presumably measured at rainfall event scale). Other aspects of stemflow, for instance the peak flux or rate of delivery of stemflow to the base of the plant, are much more likely to have been affected by rainfall intensity. I am not sure why the authors only consider overall stemflow volume, and they should make a case for neglecting other ways to characterize stemflow, including the timing of its delivery from the plant. Stemflow volume alone does not provide a complete exploration of the origin and fate of stemflow.

**Reply:**

Thank you for this comment.

As stated in this comment, the peak flux, the intensity and the rate of delivery of stemflow were indeed good indicators to characterize stemflow and explain the origin and fate of stemflow from the temporal aspects. This manuscript focused on the stemflow yield and efficiency, and their relationships with plant traits and meteorological characteristics (from P.6, Line 130 to P.7, Line 138). The indicators of $SF_b$, $SF_d$, SF%, SFP and FR were commonly used in the previous studies (Honda et al., 2015; Levia et al., 2015; Zimmermann et al., 2015; Su et al., 2016), which could provide feasible explanations to explore the bio-/abiotic influential mechanism of stemflow yield and efficiency. Actually, we have already recorded stemflow temporal dynamics, which will be interpreted in our next research.

**R1C10:** The fundamental argument of the paper is again in need of supporting evidence from the beginning of the Discussion at line 393. The authors discuss 'effective utilization' of precipitation but as pointed out above, have no data relating to this. Their data only estimate stemflow volumes on above-ground parts of the plants. How this translates to soil moisture in the root zone (allowing for evaporation and interception on litter) is not clear.

The authors should not make claims that are not supported (or supportable) using their available data. They argue in lines 404-405 about the 'effective utilization of precipitation' by the two shrub species in rainfalls of < 2 mm. However, any stemflow delivered to the base of the shrubs in what are likely to be short showers, might be largely lost to evaporation once the short event ended. This should illustrate how spurious it might be to infer utilization from stemflow data not supported by soil moisture data, or indeed by measures of transpiration by the plants. The authors proceed (e.g. line 420) to argue about energy conservation, again speculating about the utilization of stemflow from rainfall events of < 2 mm. All of this is completely unsupported by the data, and should be eliminated from the paper, or at least highlighted as completely speculative. Again, in line 430-431 the authors speculate about drought tolerance; not only do they have no supporting data, but the data that they do have were derived during the rainy season, and not in drought conditions at all. How the shrub foliage etc. might change during drought years remains unknown and the authors should eliminate all of their speculation about drought tolerance. Their data relate to stemflow alone, and they should restrict themselves primarily to discussing and interpreting those data.

Lines such as 476-478 inclusive are completely speculative, though the authors write as though they are presenting a result from their work. They refer to stemflow production under 'water stress conditions' though they did not observe this; they refer to their estimated stemflow being 'of significant importance for the survival of the xerophytic shrubs, particularly during long intervals with no rainfall' though they present absolutely no evidence to support this claim, having no data from long periods with no rainfall. All of this speculation should be eliminated from the paper, or at the very least identified as speculation not supported by any data.

Overall, the focus of the paper needs to shift from speculation to the discussion of what can validly be determined from the field evidence available, namely, the estimated stemflow volumes.

**Reply:**

Thank you for your comments and advices on some speculative discussions for the original version of this manuscript. The focus of the revised manuscript has been shifted from the addressing of some speculations to the interpreting of the measured stemflow data, and we discussed the benefits brought by higher stemflow yield and efficiency for dryland shrubs more cautiously.

To avoid confusions in this study, "precipitation utilization" has been deleted (in P.22, Line 546 and in P.24, Line 582) or changed to "employ precipitation to produce stemflow" (in P.22, Line 550 and in P.23, Line 564). Besides, we revised this manuscript carefully and tried best to guarantee the fact-based conclusions and precise expressions. The expressions of "water stress conditions" (in P.26, Line 647), "particularly during long intervals with no rainfall" (in P.26, Line 649) as described in this comment have been deleted, and "the utilization of stemflow from rainfall events of <2 mm" have been revised in P.26, Line 649.

For the better evidence-based arguments, new supporting materials have been added at the revised manuscript, including (1) new experimental data in a controlled experiment of the foliated and manually defoliated shrubs of *C. korshinskii* and *S. psammophila* during the 2015 rainy season, (2) new meteorological characteristics including wind speed, air temperature and humidity and solar radiation during the 2014 and 2015 rainy seasons, (3) new references addressing the importance of stemflow as potential resource for soil moisture replenishment at the root zone and the deep layer, and the normal functioning of dryland shrubs. Please see Reply for R1C1 for a detailed description.

**Other comments:**

Line 41: what are 'stemflow channels'? Does this imply fixed pathways?

**Reply:**

Thanks for the correcting. We have revised the "stemflow channels divert precipitation" to "stemflow delivers precipitation" in P.4, Line 57. Additionally, the verb "channel" has also been replaced by "deliver" or "transport" in P.5, Line 87 and in P.23, Line 554.

Line 41: 'pointedly' should be 'directly' or similar.
**Reply:** Done (in P.4, Line 57).

Line 44: what is meant by 'biogeochemical reactivity at the terrestrial-aquatic interface'?
**Reply:**

The "biogeochemical reactivity at the terrestrial-aquatic interface" refers to the nutrients cycling assisted by the microorganism activity while the nutrients-enriched stemflow infiltrated to the soil matrix, which was cited from the reporting of McClain et al. (2013), including total nitrogen (TN), total phosphors (TP), $NH_4^+$-N, $NO_3^-$-N, $Na^+$, $K^+$, $Ca^{2+}$, $Mg^{2+}$, $Cl^-$, $SO_4^{2-}$, etc. (Zhang et al., 2013).

For an easier understanding, this sentence had been changed to "The double-funnelling effects of stemflow and preferential flow create "hot spot" and "hot moment" by enhancing nutrients cycling rates at the surface soil matrix" in P.4, Line 60–61.

Line 58: please cite references to support the claim about 'disproportionately high influence [of stemflow] on survival and competitiveness of xerophytic shrub species'.
**Reply:** Done (in P.4, Line 72–76 and in P.24, Line 594–598).

Line 81: insert missing space before 'Murakami'.
**Reply:** Done (in P.6, Line 114–115).

Line 155: how do branches exist 'as independent individuals'?
**Reply:**

Thank you for your question. It related to the biological attributes of modular organisms. Please see Reply for R1C5 for a detailed explanation. For a better understanding, the expression of "existed as independent individuals" had been deleted at the revised manuscript (in P.9, Line 203–204).

Line 214: 'at the' should be 'in a'.
**Reply:** Done (in P.13, Line 301).

Line 238: should '4080-mm' be '40-80 mm'?
Line 475: should 'events of 12-mm' read 'events of 1-2 mm'?
Line 268 and many other instances: do not write '18-mm'; the hyphen is not allowed in the SI metric system. There must be a space between the numerical quantity and the symbol for the unit of measurement (e.g. '18 mm' is correct).
**Reply:**

Thank you for the correcting and explaining. We had corrected these errors at the revised manuscript (in P.8, Line 173, in P26, Line 650, and in P.16, Line 386).

Line 280: do the authors data justify 4 decimal places of precision? This requires fixing in many places, such as line 475.
**Reply:**
  Thank you for this comment. At the revised manuscript, we kept the fixed one decimal place of precision for all the indicator except for the SFP with the two decimal places, because SFP of one decimal place was too rough to tell a clear difference between different precipitation and BD categories.

Line 492: 'had not determined yet' should read 'have not yet been determined'.
**Reply:** This sentence had been deleted at the revised manuscript. A similar mistake had been corrected in P.6, Line 107.

**Response to Reviewer #2:**

**General reply:**

**R2C1:** This study explored stemflow yield in relations to rainfall characteristics and the plant traits of branches and leaves for two dominant shrubs (*C. korshinskii* and *S. psammophila*) during rainy seasons in the northern Loess Plateau of China. This manuscript reports important data on stemflow measurements at the scale of individual branches and highlights the effect of canopy structure (e.g. biomass, the leaf area of the branches, the leaf numbers of the branches, stemflow productivity, and the funnelling ratio) on stemflow production. The finding of this study is interesting and fall into the scope of the HESS. However, my main concern is the title, results and discussions are not really robust and can't be fully supported by data, and the interpretation is weak.

**Reply:**

Thank you for your comments and interests in this study. We have substantially revised the Title and the sections of Introduction, Materials and Methods, Results, and Discussions at the revised manuscript. Please see the detailed replies to the following comments.

**R2C2:** (1) Title: The "the effects of leaves and implications in drought tolerance" in the title is not well reflected in the results of this study. Although measurements of leaf area index (LAI), the foliage orientation, the leaf area of the branches and the leaf numbers of the branches were made in the study, results of species-specific variation of plant traits (line 236-283) just mainly qualitatively described leaf traits, branch morphology and biomass, which were not directly linked with stemflow characteristics. Moreover, results of this study indicated that precipitation amount was the most influential rainfall characteristic and stem biomass and leaf biomass were the most influential plant traits that affected stemflow in *C. korshinskii* and *S. psammophila*, so the effects of leaves on stemflow were not well investigated in this study. In the case of implications in drought tolerance, authors mainly discussed with personal speculations, there were not solid soil water data to verify it. So I suggest author could delete "the effects of leaves and implications in drought tolerance" from the title.

**Reply:**

Thank you for your comments and advices regarding the title of this manuscript.

We had revised the title as "Comparisons of stemflow and its bio-/abiotic influential factors between two xerophytic shrub species" (please see P.1, Title).

The effects of leaves on stemflow has been further interpreted with a controlled experiment of comparing stemflow yield between the foliated and manually defoliated shrubs during the rainy season (in P.11, Line 235–251, in P.18, Line 433–447, in P.31, Line 758–778, in P.33, Line 812–820, and in P.50, Line 1107–1110).

Some speculation, such as "drought tolerance" has been deleted from the title and other places in P.2, Line 39, in P.7, Line 140, in P.8, Line 166, in P.24, Line 601–603, and in P.32, Line 804. Please see the detailed description at Reply to R1C10 at Response to Reviewer #1.

**R2C3:** (2) Introduction: The objectives of this study were not clear, what's the new findings made by this study? What's the knowledge gaps in stemflow researches for shrubs? In fact, stemflow of *C. korshinskii* and *S. psammophila* were already studied in China, what's the difference between studies? I wonder if authors can highlight the stemflow yield from branches and stemflow productivity between shrubs.

**Reply:**

Thank you for your comments and constructive advices regarding the new findings of this manuscript, which were listed as follow.

1) We introduced the indicator of stemflow productivity (Yuan et al., 2016) and assessed stemflow efficiency for the first time with the combined results of funnelling ratio and stemflow productivity in this study (in P.2, Line 26). Along with other indicators of $SF_b$, $SF_d$ and SF%, the inter- and intra-specific differences of stemflow yield and efficiency were studied comprehensively at this manuscript (as indicated at the following Table R2-1).

2) We studied the effects of meteorological characteristics and plant traits particularly at the finer leaf scale affecting stemflow yield and efficiency.

A direct evidence regarding leaf's effects on stemflow yield was provided at this manuscript with a controlled experiment of comparing the branch stemflow yield ($SF_b$) between the foliated and manually defoliated *C. korshinskii* and *S. psammophila* during the 2015 rainy season. In relative to the previous studies, it was believed the first controlled experiment at field, which guarantee the identical stand conditions and meteorological characteristics (as indicated at the following Table R2-2). We found that the newly exposed branch surface at the defoliated period and the resulting rainfall intercepting effect might be of significance for stemflow production, which was generally ignored by previous studies.

Table R2-1. Comparison of the advantage and drawback between stemflow indicators.

| NO. | Stemflow indicators | Expressions | Advantages | Drawbacks |
|---|---|---|---|---|
| 1 | Stemflow volume ($SF_v$, mL) | N/A | Simple and clear to present stemflow yield. | Hard to compare the $SF_b$-specific differences because of the huge variation of plant traits between |
| 2 | Stemflow equivalent water depth ($SF_d$, mm) | $SF_d = SF_v/CA$ | | |
| 3 | Stemflow percentage of | SF% = $SF_d/P$ | | |

| | | | | |
|---|---|---|---|---|
| | incident precipitation (SF%, %) | | | different plant functional types. |
| 4 | Funneling ratio (FR) | $FR = SF_v/(P*S)$ | 1) Available to compare inter-specific stemflow efficiency; 2) Commonly used to evaluate stemflow efficiency. | Relative a weak connection with plant growth, e.g., biomass accumulation and allocating patterns. |
| 5 | Stemflow productivity (SFP, mL·g$^{-1}$) | $SFP = SF_v/BMB$ | Characterizing stemflow efficiency and relating closely with biomass accumulating and allocating. | No response to variation of meteorological characteristics. |

Note: CA is the canopy area; P is the precipitation amount; and BMB is the branch biomass.

Table R2-2. Previous studies regarding leaf's effects on stemflow by comparing stemflow yield at the foliated and defoliated period.

| The effects of leaves on stemflow yield | Relevant studies | Reference |
|---|---|---|
| Negative effects | Oak forest in Holland | Dolman, 1987 |
| | Oak forest in Spain | Muzylo et al., 2009 |
| | Laurel forest in Japan | Masukata et al., 1990 |
| | Beech plantation in England | Neal et al., 1993 |
| Positive effects | Stewartia forest in Japan | Liang et al., 2009 |
| Neglectable effects | Desert shrubs in USA | Martinez-Meza and Whitford, 1996 |
| | Broad-leaves forest in Japan | Deguchi et al., 2006 |

**R2C4:** (3) Materials and Methods: As shrubs grow during the rainy period, at what period (time) or measurement frequency do authors measure plant traits, particularly for biomass (line 175), how can you confirm them represent real plant trait dynamics, which were not clearly described in the text. Line 155: what's the "modular organisms and multi-stemmed shrub"?
**Reply:**

Thank you for your comments on experimental design of this manuscript.

It is a good question regarding the time dependency of plant traits measurements, particularly for biomass. We measured biomass and leaf traits simultaneously at middle August when the shrubs showed maximum vegetative growth during the rainy season (in P.12, Line 262). If conducting the dynamic measurements, the shrubs would be constantly disturbed even destroyed, and the results of stemflow yield and efficiency would be biased in this study. The variation of those plant traits was small during the experimental period, and they were generally ignored (Siles et al., 2010a, b; Levia, et al., 2015; Zhang et al., 2015).

The modular organism are those organisms, whose zygote develops into a discrete unit (module), and then produces more units like itself, rather than developing into a complete organism (Allaby, 2010). Each module seeks its own survival goals and the resulting organism level behavior is not centrally controlled (Firn, 2004) (in P.9, Line 202–205). The multi-stemmed shrubs have no trunk but have multiple branches that radiate from their base (in P.8, Line 167–169), e.g., *C. korshinskii* and *S. psammophila* in this study. These two shrub species are the ideal experimental objects to study stemflow at the branch scale.

**R2C5:** (4) Results: For the most part of the "3.1 Species-specific variation of plant traits", it is not really the results of the study, I would suggest authors move some of the description of *C. korshinskii* and *S. psammophila* to the section of "Materials and Methods". Line 387-390: it is not clear, why big difference existed between rains 10 mm and the heavy rain.
Reply:

Thank you for your comments. The description of plant traits of *C. korshinskii* and *S. psammophila* has been moved to the "Materials and Methods" section as required in P.8, Line 169–175.

We have discussed the reasons for different plant trait of leaves and branches affecting SFP between smaller rains ≤10 mm and heavier rains >15 mm, respectively. It might relate to the specific stemflow producing processes during different-sized rains. Please see the detailed description in P.22, Line 529–535.

**R2C6:** (5) Discussions: I would suggest authors focus on the interpretation of the results of this study, but not speculations on utilization of more rains via a low precipitation, there was not direct evidence or robust data to support the proposed conclusion.
**Reply:**

Thank you for your comments on interpreting the results of this manuscript.

The focus of the revised manuscript has been shifted from the discussing of some speculations to the interpreting of the measured stemflow data. We have deleted the vague expressions of "water stress conditions" (in P.26, Line 647), "particularly during long intervals with no rainfall" (in P.26, Line 649). The phrase of "implication in drought tolerance" has also been deleted in the title (in P1, the Title). To avoid confusions at this manuscript, "precipitation utilization" has been deleted (in P.22, Line 546, and in P.24, Line 582) or changed to "employ precipitation to produce stemflow" (in P.22, Line 550, and in P.23, Line 564). More detailed description please see Reply to R1C1and Reply to R1C10 at the Response to reviewer #1.

**R2C7:** (6) English languages needs refine by a native English speakers.

**Reply:**

  Thank you for this comment. We have already sent this manuscript for a professional language editing. Please see the certificate as follow. Furthermore, the language of revised manuscript has been double checked.

[Figure]

**Nature Research Editing Service Certification**

This is to certify that the manuscript titled Greater stemflow yield and efficiency of Caragana korshinskii than Salix psammophila: leaf's effect and implication for drought tolerance was edited for English language usage, grammar, spelling and punctuation by one or more native English-speaking editors at Nature Research Editing Service. The editors focused on correcting improper language and rephrasing awkward sentences, using their scientific training to point out passages that were confusing or vague. Every effort has been made to ensure that neither the research content nor the authors' intentions were altered in any way during the editing process.

Documents receiving this certification should be English-ready for publication; however, please note that the author has the ability to accept or reject our suggestions and changes. To verify the final edited version, please visit our verification page. If you have any questions or concerns over this edited document, please contact Nature Research Editing Service at languageediting@as.springernature.com.

| | |
|---|---|
| **Manuscript title:** | Greater stemflow yield and efficiency of Caragana korshinskii than Salix psammophila: leaf's effect and implication for drought tolerance |
| **Authors:** | Chuan Yuan, Guangyao Gao, Bojie Fu |
| **Key:** | EDA6-0499-5F73-CC0C-4063 |

This certificate may be verified at **secure.authorservices.springernature.com/certificate**.

Nature Research Editing Service is a service from Springer Nature, one of the world's leading research, educational and professional publishers. We have been a reliable provider of high-quality editing since 2008.

Nature Research Editing Service comprises a network of more than 900 language editors with a range of academic backgrounds. All our language editors are native English speakers and must meet strict selection criteria. We require that each editor has completed or is completing a Masters, Ph.D. or M.D. qualification, is affiliated with a top US university or research institute, and has undergone substantial editing training. To ensure we can meet the needs of researchers in a broad range of fields, we continually recruit editors to represent growing and new disciplines.

Uploaded manuscripts are reviewed by an editor with a relevant academic background. Our senior editors also quality-assess each edited manuscript before it is returned to the author to ensure that our high standards are maintained.

CERTIFICATE EDA6-0499-5F73-CC0C-4063 - AUGUST 10, 2016 - PAGE 1 OF 1

**Fig. R2-1.** The certificate for language editing.

If you have any further questions about this revision, please contact us.

Sincerely Yours,

Dr. Guangyao Gao (gygao@rcees.ac.cn)

[revised manuscript text omitted]

---

## Author Response (AR2)

**中国科学院生态环境研究中心**
**Research Center for Eco-Environmental Science**
**Chinese Academy of Sciences (CAS)**

北京市海淀区双清路 18 号 邮编：100085  Website: www.rcees.ac.cn
**18 Shuangqing Road, Haidian District, Beijing 100085, P. R. China**
Tel: 0086-10-62911239   Email: gygao@rcees.ac.cn

February 17, 2017

Memorandum

To:   Prof. Dr. Lixin Wang, Editor of *Hydrology and Earth System Science*

Subject: **Revised Version #2 of hess-2016-420**

Dear Prof. Dr. Wang,

  We have substantially revised our manuscript entitled as "*Comparisons of stemflow and its bio-/abiotic influential factors between two xerophytic shrub species*" after considering all the comments of Prof. Dr. David Dunkerley and another anonymous reviewer, which are of great help to improve this manuscript.

  The following are the point-to-point response to all these comments, including (1) Response to Reviewer #1 (Prof. Dr. David Dunkerley), (2) Response to the anonymous Reviewer #2, and (3) The marked-up manuscript version, respectively.

**Response to Reviewer #1, Prof. Dr. David Dunkerley:**

**R1C1:** This manuscript is somewhat improved over the version that I read previously. The paper makes it clear that the authors carried out a significant amount of work, both in the field, and in data processing and analysis.
However, I do have some concerns about the work done, and about the way in which it has been written up.

**Reply:**

We really appreciated Prof. Dunkerley for the comments and suggestions, which were of great help to improve the overall quality of this manuscript, particularly in the rigorousness of experiment design, results interpretation and English expression. The version #2 of this manuscript had been revised carefully and addressed all the concerns, and we tried best to upload a qualified manuscript as required.

**R1C2:** In terms of the work done, I have concerns about the stemflow collecting system used by the authors. Despite their assurances, the photographs supplied with the paper show quite wide gaps between the stem and the stemflow collar, that may have allowed rain or released throughfall from branches above to enter the stemflow collars and be erroneously counted as stemflow. I would like to have seen the authors make some estimate of how large this error could be.

**Reply:**

Thank you for commenting on the measuring errors of stemflow yield originated from the field experimental setting.

As mentioned in this comment, the stemflow yield might be indeed overestimated in this study, which was affected by the precipitation and throughfall. However, that might be unavoidable, especially at the field conditions. Therefore, we took various experimental techniques to mitigate the experimental errors between the measuring values and the real values, including the stemflow collecting method of fossil collars (rather than spiral tubes) (in P.9, Lines 216–217), the collar installation positions (the lower part of branches at the canopy outskirt) (in P.10, Lines 217–221), the limited collar diameter (in P.10, Lines 221–223), the periodical checking against leakage and blockage (in P.10, Lines 223–224). Nevertheless, it was difficult to estimate exactly how large the measuring errors could be, considering the sporadic distribution and intensity of the precipitation and throughfall. To perform an objective analysis, we made a special statement to describe the possible overestimation of stemflow in this manuscript in P.10, Lines 228–234.

**R1C3:** The authors still say nothing about wind speed or about the extent to which the rain

had an oblique approach angle. They seem overly concerned about leaf architecture and not sufficiently concerned about the possible effects of oblique, wind-driven rain on the field measurements. Likewise, what was the effect of wind in dislodging drops that might otherwise have become stemflow? Considerations of this kind cause me to wish that the authors could be somewhat more cautious in their conclusions, and at least acknowledge the potential effects of variables that they do not consider in their analysis.

**Reply:**

Thank you for commenting on the inadequate consideration of abiotic influences affecting stemflow in this study. At the revised version #2 of this manuscript, we supplemented raindrop attributes, including the average raindrops diameter (D, mm), the terminal velocity of raindrops (V, $m \cdot s^{-1}$), and raindrops inclination angle (A, °) (in P.8–9, Lines 191–195, P.12, Lines 271–279 and P.15, Lines 351–352), and revised our conclusions in a more cautious manner. As shown at Table R1C3 bellow, there was no significant correlations of stemflow yield ($SF_b$) with D, A and V indicated by the Pearson correlation analysis, which was further confirmed by the following Partial correlation analysis. In contrast to the precipitation amount, there might be much weaker effects of raindrops size, velocity and inclination angle on stemflow yield of *C. korshinskii* and *S. psammophila* in this study. Please see the detailed description of biotic influential mechanism of stemflow yield in P.20, Lines 500–505.

**Table R1C3.** Pearson correlation between raindrop attributes and $SF_b$ at different BD categories

| Species | BD categories (mm) | Analysis Parameters | D (mm) | V ($m \cdot s^{-1}$) | A (°) |
|---|---|---|---|---|---|
| *C. korshinskii* | 5–10 | Correlation | 0.31 | 0.36* | -0.03 |
| | | Sig. | 0.07 | 0.03 | 0.85 |
| | | N | 36 | 36 | 36 |
| | 10–15 | Correlation | 0.25 | 0.31 | -0.06 |
| | | Sig. | 0.14 | 0.07 | 0.74 |
| | | N | 36 | 36 | 36 |
| | 15–18 | Correlation | 0.27 | 0.32 | -0.02 |
| | | Sig. | 0.11 | 0.06 | 0.89 |
| | | N | 36 | 36 | 36 |
| | 18–22 | Correlation | 0.25 | 0.31 | 0.01 |
| | | Sig. | 0.14 | 0.07 | 0.98 |
| | | N | 36 | 36 | 36 |
| | Avg.(BD) | Correlation | 0.27 | 0.32 | -0.03 |
| | | Sig. | 0.12 | 0.06 | 0.87 |
| | | N | 36 | 36 | 36 |
| *S. psammophila* | 5–10 | Correlation | 0.27 | 0.32 | -0.04 |
| | | Sig. | 0.11 | 0.06 | 0.81 |
| | | N | 36 | 36 | 36 |

[Figure]

| | | | | |
|---|---|---|---|---|
| | Correlation | 0.29 | 0.34* | -0.01 |
| 10–15 | Sig. | 0.09 | 0.04 | 0.97 |
| | N | 36 | 36 | 36 |
| | Correlation | 0.32 | 0.37* | -0.01 |
| 15–18 | Sig. | 0.05 | 0.03 | 0.99 |
| | N | 36 | 36 | 36 |
| | Correlation | 0.36* | 0.41* | 0.02 |
| 18–22 | Sig. | 0.03 | 0.01 | 0.91 |
| | N | 36 | 36 | 36 |
| | Correlation | 0.33* | 0.38* | 0.01 |
| Avg.(BD) | Sig. | 0.05 | 0.02 | 0.98 |
| | N | 36 | 36 | 36 |

Note: D, V and A are the diameter, velocity and inclined angle of raindrops; * correlation is at the 0.05 level.

**R1C4:** The authors have been more circumspect following my previous review, and have backed away from claiming that stemflow is critical to drought survival, now stating that it 'might be' important. However, they are still in my view insufficiently careful with their argument. For instance, on page 3, in their Introduction, the authors claim that 'Stemflow delivers precipitation directly into the root zone…' (line 42). But of course this is often not the case, and instead the stemflow arrives at a litter layer beneath the plant where, in addition, the soil is sometimes hydrophobic. The authors need to be more careful (especially as they have no evidence of soil moisture changes caused by stemflow) in making claims of this kind. Some fraction of the stemflow likely reaches the root zone, but just how much does so is an important question that should not be overlooked. The authors also become rather enthusiastic in line 531, where they argue that efficient stemflow collection might be of 'great' importance – or perhaps this should be just of 'some' importance, until we have some actual evidence.

**Reply:**

Thank you for this comment. At the version #2 of this manuscript, we double-checked the description and interpretation of analysis results, thus demonstrating the ecological effects of stemflow more objectively and cautiously. For instance, the description of "stemflow delivers precipitation directly into the root zone …" had been revised to "stemflow delivers precipitation to the plant root zone more efficiently via preferential root paths, worm paths and soil macropores …" in P.3, Lines 42–44. The statement of "That meant a lot for xerophytic shrubs particularly during the rainy season" had been revised to "But during lighter rains, the larger amount stemflow produced in more efficient manner might benefit xerophytic shrubs, for more soil moisture could be recharged especially at the root zone" in P.24, Line 591–593. The description of "*C. korshinskii* produced stemflow with a greater amount in a more efficient manner might be of great importance in employing precipitation to acquire water (Murakami, 2009) at dryland ecosystems" had been deleted, and we put forward a suggestion for the future study that "… in addition to quantify the soil moisture recharge, a thorough study was required to depict the stemflow infiltration process, particularly at the water-stressed environment" in P.24, Lines 593–595.

**R1C5:** Given the very large literature on stemflow, I also think that the authors are too sweeping on page 3 where they claim (line 64) that '..previous studies have usually ignored stemflow…'. This would be news to the authors of hundreds of papers on stemflow.
**Reply:**
Thank you for this comment. We have revised the confusing statement of "…previous studies have usually ignored stemflow…" to "some studies neglected the dynamics of stemflow yield by setting a fixed percentage of incident precipitation in the range of 1%–8% (Dykes, 1997; Germer et al., 2006; Hagyó et al., 2006), even ignored stemflow while computing water balance of terrestrial ecosystem (Llorens and Domingo, 2007; Zhang et al., 2016), which underestimated its disproportionately high influence on xerophytic shrub species (Andersson, 1991; Levia and Frost, 2003; Li, 2011)" in P.3–4, Lines 65–70.

**R1C6:** Written English is still poor in places. Especially in section 2.1 (study area), the authors use past tense inappropriately (e.g. line 145 should say that the shrubs 'are' multi-stemmed, not 'were'; this error occurs repeatedly in the whole top half of page 7). In general, 'grew' should be 'grow', 'was' should be 'are', and so on. The authors still use the SI metric system carelessly. For instance, '20-cm-diameter' (line 167) is incorrect, and should be '20 cm diameter' ; similar errors occur in line 204, line 417, and elsewhere.
**Reply:**
Thanks for this comment to further improve the written English of this manuscript. The inappropriate verb tense has been corrected particularly in *Section 2.1 Study area* in P.6, Line 139 and P.7–8, Lines 158–170. Other incorrect expressions in English were also rectified, such as the definite article in P.3, Line 51 and in P.18, Line 431, the comparative adjectives in P.2, Line 37, the singular and plural nouns in P.8, Line 187.

Besides, we revised the inappropriate and nonuniform expressions of SI metric system, including the "20-cm-diameter" to "20 cm diameter" in P.8, Line 180, the "0.5-cm-diameter" to "0.5 cm diameter" in P.10, Line 223, the ">18-mm branches" to ">18 mm branches" in P.19, Line 468, the "rain-fed dryland ecosystems" to "rainfed dryland ecosystems" in P.3, Line 50 and Line 60, the "eco-hydrological flux" to "ecohydrological flux" in P.3, Line 56, the "eco-hydrological processes" to "ecohydrological processes" in P.6, Line 128, the "eco-hydrological perspective" to "ecohydrological perspective" in P.6, Line 132, the "eco-zone" to "ecozone" in P.4, Line 76, the "different-sized branches" to "different sized branches" in P.4, Line 86, in P.13, Line 306, and in P.29, Line 727, the "wind-proofing and dune-stabilizing" to "wind proofing and dune stabilizing" in P.7, Line 156, the "inverted-cone canopy" to "inverted cone canopy" in P.7, Line 158, the "lanceolate-shaped" to "lanceolate shaped" in P.7, Line 161, the "strip-shaped leaves" to "strip shaped leaves" in P.7, Line 164,

[Figure]

**中国科学院生态环境研究中心**
**Research Center for Eco-Environmental Science**
**Chinese Academy of Sciences (CAS)**

北京市海淀区双清路 18 号 邮编：100085  Website: www.rcees.ac.cn
**18 Shuangqing Road, Haidian District, Beijing 100085, P. R. China**
Tel: 0086-10-62911239   Email: gygao@rcees.ac.cn the "step-wise regression" to "stepwise regression" in P.20, Line 502, and the "semi-arid" to "semiarid" in P.26, Line 642.

**R1C7:** The authors need to proof-read their work carefully. For instance, they often refer to 'defoliated and manually defoliated shrubs' (e.g. line 112, and again in lines 117-118), when they mean 'foliated and manually defoliated'. This becomes quite confusing. The authors sometimes refer to the shrubs competing for 'lights' (e.g. line 181, line 520) but this should simply be 'light'.

**Reply:**

Thanks for this comment. As required, we have corrected these mistakes, such as the "defoliated" in P.6, Line 121 and Lines 125–126, and the "lights" in P.9, Line198, in P.23, Line 576, and in P.30, Line 745.

**R1C8:** My overall feeling about the paper is that the work is generally of good standard, but that the authors should again check their paper carefully for errors; give thought and comment to the quality of the field data (e.g. was throughfall counted as stemflow?). At the same time, if possible, the manuscript should be shortened as the Discussion in particular is quite long and a little repetitive.

**Reply:**

We appreciated Prof. Dunkerley for the comments and suggestions, which were of great help to improve this manuscript. As required, we had rectified the incorrect expressions in English and SI metric system (see Reply to R1C6), corrected the clerical errors (see Reply to R1C7), and revised some imprecise and sweeping claims (see Reply to R1C4 and R1C5). We also stated the limitation of this research in the measuring errors (see Reply to R1C2), and the flaws of the controlled field experiment (see the following Reply to R2C1). Furthermore, some issues have been put forward for future studies in P.6, 127–129, in P.10, Lines 232–234, in P.29, Lines 719–723.

Besides, some repetitive content has been removed particularly in the *Discussion* section of this manuscript in P.22, Lines 539–542 and Lines 547–552, in P.23, Lines 567–572 and Lines 574–578, in P.23–24, Lines 580–586, in P.24, Lines 587–589, in P.25, Lines 611–628, and in P. 28, Lines 689–695.

If you have any further questions about this revision, please contact us.

Sincerely Yours,

Dr. Guangyao Gao (gygao@rcees.ac.cn)

[Figure]

**中国科学院生态环境研究中心**
**Research Center for Eco-Environmental Science**
**Chinese Academy of Sciences (CAS)**
北京市海淀区双清路 18 号 邮编：100085  Website: www.rcees.ac.cn
**18 Shuangqing Road, Haidian District, Beijing 100085, P. R. China**
Tel: 0086-10-62911239   Email: gygao@rcees.ac.cn

**18 Shuangqing Road, Haidian District, Beijing 100085, P. R. China**

[revised manuscript text omitted]